# MV-FGAD: Towards Efficient and Effective Federated Graph Anomaly Detection via Multi-view Learning

**Junyi Yan**[1] **Ke Liang**[1] **Hao Yu**[1] **Meng Liu**[2] **Hao Tan**[1] **Tianrui Liu**[1] **Jun-Jie Huang**[1] **Xinwang Liu**[1]

## Abstract

Federated graph anomaly detection (GAD) aims to identify abnormal nodes in distributed subgraphs through federated learning. However, existing methods suffer from two limitations. 1) Their reliance on neighborhood aggregation assumes that anomalous information can be sufficiently captured, which often fails in federated learning with partitioned client subgraphs. 2) They overlook the detection bottleneck caused by weak attribute or structural anomalies. To tackle these challenges, we revisit federated GAD and reveal that weak anomalies exhibit harder-to-detect signals compared to strong anomalies. Specifically, we propose MV-FGAD, an efficient and effective federated GAD framework for mining anomalies of varying strengths. MV-FGAD introduces a federated knowledge learning module to aggregate and broadcast shared knowledge, which is further exploited to optimize local topological structures. Moreover, it designs a multi-view learning mechanism to capture diverse anomaly patterns, and adopts Mahalanobis distance–based scoring strategy to quantify node abnormality across views. Extensive experiments on real-world datasets of varying types and scales demonstrate MV-FGAD's efficiency and effectiveness. Our code is publicly available at https://github.com/Junyi-Yan/MV-FGAD.

## 1. Introduction

Graph anomaly detection (GAD) aims to identify abnormal instances (e.g., nodes, edges, subgraphs and graphs)

[1]College of Computer Science and Technology, National University of Defense Technology, Changsha, 410073, China [2]School of Artificial Intelligence, Henan University, Zhengzhou, 450046, China. Correspondence to: Tianrui Liu <trliu@nudt.edu.cn>, Xinwang Liu <xinwangliu@nudt.edu.cn>.

*Proceedings of the 43rd International Conference on Machine Learning*, Seoul, South Korea. PMLR 306, 2026. Copyright 2026 by the author(s).

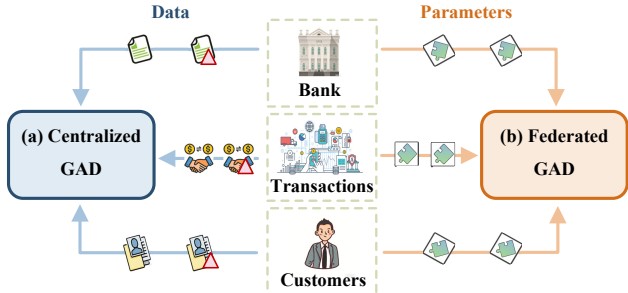

*Figure 1.* A toy example illustrating (a) Centralized GAD and (b) Federated GAD, with anomalous data indicated by triangles.

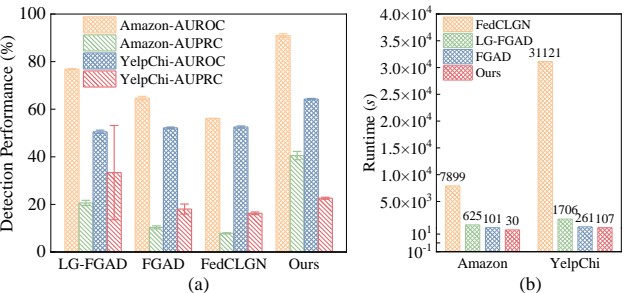

*Figure 2.* Comparison of efficiency and effectiveness between existing methods and ours: (a) Performance and (b) Runtime on Amazon-all and YelpChi.

that deviate significantly from normal patterns by jointly modeling node attributes and graph topology (Ma et al., 2021; Qiao et al., 2025b), and has been widely applied in domains such as fraud detection and cybersecurity (Wang et al., 2025b; Zhang et al., 2022; Yan et al., 2023a; Wang & Zhu, 2022; Pan et al., 2025). Although existing centralized GAD methods have achieved promising performance (Ding et al., 2021; Duan et al., 2023a;b; Ma et al., 2024; Liu et al., 2024c; Gao et al., 2024; Yan et al., 2025; Qiao et al., 2025a; Yan et al., 2023b), they require aggregating all data for centralized training, which raises serious privacy concerns and limits real-world deployment. As shown in Fig. 1a, financial transaction systems contain sensitive information, including bank records, transactions and personal data. Directly sharing such data poses substantial security risks and may even facilitate financial fraud. Therefore, developing a privacy-preserving collaborative GAD framework is essential for improving security.

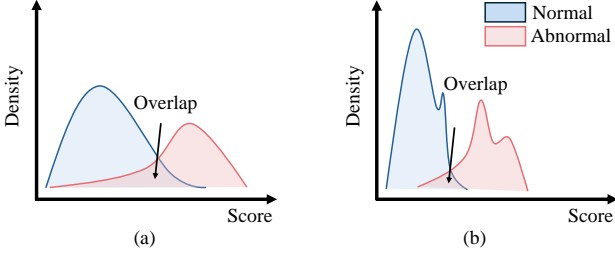

*Figure 3.* Analysis of the impact of weak anomalies on anomaly score distribution. (a) Results without mining weak anomalies. (b) Ideal results after mining weak anomalies.

Federated graph learning (FGL) combines federated learning (FL) with graph neural networks (GNNs) to enable privacy-preserving collaborative training on graph-structured data (Liu et al., 2025e; Yu et al., 2025a; Liu et al., 2025b; Liang et al., 2024a; Liu et al., 2024a; Ju et al., 2025; Tan et al., 2025; Huang et al., 2024). As shown in Fig. 1b, federated graph anomaly detection (federated GAD)[1] allows multiple clients to collaboratively train models by exchanging model parameters without sharing data, thereby alleviating the privacy risks of centralized GAD.

Representative federated GAD methods include LG-FGAD (Cai et al., 2024a), FGAD (Cai et al., 2024b) and FedCLGN (Wu et al., 2025). However, they generally assume that anomalies can be effectively characterized by rich neighborhood information, which often breaks down in FL due to incomplete local neighborhoods. Moreover, they do not explicitly address detection bottlenecks caused by weak attribute or structural anomalies. Experiments on the Amazon-all and YelpChi datasets in Fig. 2 further indicate that although LG-FGAD achieves competitive detection performance, its large variance reflects unstable effectiveness. In addition, existing methods incur substantial runtime, which limits efficiency.

To address these challenges, we revisit federated GAD, focusing on anomalies of varying strengths. We observe that weak attribute or structural anomalies are difficult to capture, which blurs the learned boundary between normal and abnormal nodes and results in overlapping anomaly score distributions with indistinct density peaks, as shown in Fig. 3a, while Fig. 3b depicts the ideal results. As a result, relying on a single aggregation strategy or learning view is insufficient to effectively mine anomalous signals of varying strengths. Notably, we find that multi-view learning can leverage complementary information from multiple perspectives (Liu et al., 2025c;d; Feng et al., 2025; Xu et al., 2024; Guan et al., 2025; 2026; Liu et al., 2025a), providing a powerful solution for mining anomaly patterns of different strengths and types.

---

[1] In this paper, we focus on node-level federated GAD and refer to it as "federated GAD".

In this paper, we propose MV-FGAD, an efficient and effective federated GAD framework based on multi-view learning. Specifically, we introduce a federated knowledge learning module that leverages standard federated collaborative training to aggregate local knowledge, while enabling cross-client knowledge sharing through federated knowledge transfer. Subsequently, to improve effectiveness, we design a multi-view learning mechanism tailored to anomalies of varying strengths and types, enabling the model to comprehensively capture diverse anomalous signals. Finally, to achieve efficient anomaly scoring, we adopt a Mahalanobis distance (MHD)-based scoring module to quantify the deviation of node representations learned from different views, and compute the final anomaly score of each node by averaging the MHDs across all views. In summary, our contributions can be summarized as follows:

- **Problem:** To overcome the inefficiency and ineffectiveness of existing methods that rely on neighborhood aggregation, we revisit federated GAD by directly targeting anomaly mining of varying strengths.

- **Algorithm:** We propose MV-FGAD, an efficient and effective federated GAD framework, which designs multi-view learning to identify anomalies induced by diverse anomaly patterns across clients and measures node-wise deviation using the MHD.

- **Experimental Findings:** We conduct extensive experiments to evaluate the efficiency and effectiveness of MV-FGAD on seven real-world benchmark datasets. Specifically, comprehensive analyses are provided on the effectiveness, efficiency and other aspects, including robustness.

## 2. Related Work

### 2.1. Federated Graph Learning (FGL)

With the rapid development of data mining and graph learning techniques (Liu et al., 2026; Yu et al., 2025b; Huang et al., 2025; Song et al., 2026), FGL has attracted increasing attention in recent years. FGL aims to train GNNs with improved performance for distributed graph data (Yu et al., 2024; Huang et al., 2023). Unlike traditional FL, FGL accounts for graph topology during training (Liu et al., 2024b; Yu et al., 2025a). In general, FGL can be categorized into graph-level and node-level according to the type of graph data on clients. For graph-level FGL, each graph is treated as an individual sample, such as molecular graphs, and each client owns a collection of graphs. FedStar (Tan et al., 2023) shares structural encodings while preserving personalized node features. FedSSP (Tan et al., 2024) mitigates knowledge conflicts by sharing common spectral knowledge and allows personalized graph adaptation for

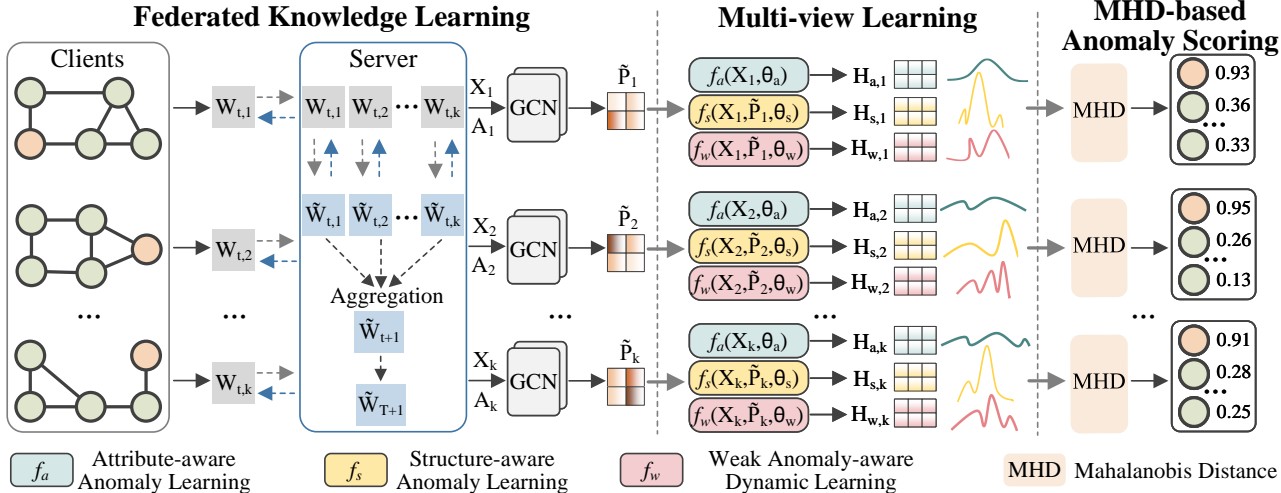

*Figure 4.* The overall framework MV-FGAD. Firstly, federated knowledge learning conducts collaborative training to obtain shared federated knowledge $\widetilde{\mathbf{W}}_{T+1}$, which is used to optimize local topology and derive the probability propagation matrix $\widetilde{\mathbf{P}}$. Then, multi-view learning captures diverse anomaly patterns via attribute-aware anomaly learning, structure-aware anomaly learning and weak anomaly-aware dynamic learning. Finally, each client applies MHD-based anomaly scoring to measure node deviations across all views.

each client. GCFL (Xie et al., 2021) dynamically clusters clients based on GNN gradients to reduce structural and feature heterogeneity across client graphs. For node-level FGL, each node in the graph is regarded as a sample, such as social networks, and each client stores a subgraph. FGGP (Wan et al., 2024) uses clustering prototypes to enhance prediction performance for multi-domain data across clients. FedGCDR (Yang et al., 2024) ensures privacy-preserving knowledge transfer while filtering harmful information via graph expansion, target-domain training, and fine-tuning. GraphFL (Wang et al., 2022) enables rapid cross-domain adaptation through model-agnostic meta-learning combined with self-training.

## 2.2. Graph Anomaly Detection (GAD)

GAD aims to identify anomalous instances that deviate from the majority of normal instances (nodes, edges, subgraphs and graphs) (Kim et al., 2022; Qiao et al., 2025b). In this work, we focus on node-level GAD. GAD methods can be categorized into centralized and federated based on the training paradigm. **Centralized GAD** detects anomalies by sharing and utilizing graph data with complete neighborhood information. CoLA (Liu et al., 2021) exploits contrastive self-supervised learning by "target node v.s. local subgraph" to capture anomaly patterns. GAD-NR (Roy et al., 2024) reconstructs local structure, self-attributes and neighbor attributes based on node representations, and distinguishes anomaly through neighborhood reconstruction loss. STIM (Yan et al., 2025) addresses message passing bottleneck caused by anomalies at critical crossroads via neighborhood structure optimization. To effectively address the challenges faced by centralized GAD across data silos, **federated GAD**

is proposed to indirectly train anomaly detection capability through collaborative parameter optimization across clients. LG-FGAD (Cai et al., 2024a) adopts local–global strategy by maximizing and minimizing mutual information to improve detection performance. FedCAD (Kong et al., 2024) employs anomaly-aware updating strategy that encourages clients to focus on aggregating anomalous neighbor representations from other clients. FGAD (Cai et al., 2024b) generates self-perturbed graphs and applies a student–teacher knowledge distillation framework with federated learning. FedGAD (Fang et al., 2025) utilizes multi-scale contrastive objectives to learn rich representations. FedCLGN (Wu et al., 2025) maintains a global pool of negative sample pairs to optimize contrastive learning. Despite achieving promising results, their reliance on neighborhood aggregation often fails to effectively capture anomalous patterns, and they overlook the mining of weak anomaly signals. As a result, these methods struggle to clearly delineate the boundary between normal and abnormal nodes within client subgraphs. Motivated by these, we explore an efficient and effective federated GAD framework.

## 3. Problem Statement

**Notations:** Given $\mathcal{G} = \{\mathcal{G}_1, \ldots, \mathcal{G}_M\}$ denotes a graph dataset which consists of $M$ subgraphs. For each subgraph $\mathcal{G}_m = (\mathcal{V}_m, \mathcal{E}_m, \mathbf{X}_m)$, $\mathcal{V}_m = \{v_1, \cdots, v_n\}$ denotes the set of nodes and $\mathcal{E}_m \subseteq \mathcal{V}_m \times \mathcal{V}_m$ denotes the set of edges, $\mathbf{X}_m \in \mathbb{R}^{n \times d}$ denotes the node attribute matrix, with $\mathbf{x}_i \in \mathbb{R}^d$ representing the attribute vector of node $v_i$. The graph structure is represented by the adjacency matrix $\mathbf{A}_m \in \mathbb{R}^{n \times n}$, where $a_{ij} \in [0, 1]$ denotes the edge

weight between nodes $v_i$ and $v_j$. Besides, the label matrix is $\mathbf{Y}_m = \{y_1, \ldots, y_n\}$.

**Federated Learning Client Subgraph:** Given the FL system consisting of one server and $M$ clients. The graph $\mathcal{G} = (\mathcal{V}, \mathcal{E}, \mathbf{X})$ is partitioned into $M$ subgraphs based on the Louvain algorithm (Blondel et al., 2008), where $\mathcal{G}_i = (\mathcal{V}_i, \mathcal{E}_i, \mathbf{X}_i)$ is the subgraph of client $C_i$. Note that $\forall\, i \neq j,\ \mathcal{V}_i \cap \mathcal{V}_j = \varnothing$.

**Graph Neural Networks (GNNs):** GNNs learn representations through iterative neighborhood aggregation and message passing. In this work, we design MV-FGAD based on the graph convolutional network (GCN) (Kipf, 2017), whose layer-wise forward propagation can be defined as follows:

$$\mathbf{H}^{(l+1)} = \sigma\left( \widetilde{\mathbf{D}}^{-\frac{1}{2}} \widetilde{\mathbf{A}} \widetilde{\mathbf{D}}^{-\frac{1}{2}} \mathbf{H}^{(l)} \mathbf{W}^{(l)} \right), \qquad (1)$$

where $\mathbf{H}^{(l+1)}$ and $\mathbf{H}^{(l)}$ are the hidden representation matrices at the $(l+1)$-th and $l$-th layers, respectively, with $\mathbf{H}^{(0)} = \mathbf{X}$. Here, $\widetilde{\mathbf{A}} = \mathbf{A} + \mathbf{I}$ denotes the adjacency matrix with self-loops, $\widetilde{\mathbf{D}}$ is the corresponding degree matrix, $\mathbf{W}^{(l)}$ is the trainable weights of the $l$-th layer, and $\sigma$ is a nonlinear activation function.

**Semi-supervised Federated GAD:** For the client $C_m$ with graph $\mathcal{G}_m = (\mathcal{V}_m, \mathcal{E}_m, \mathbf{X}_m)$, the nodes in unlabeled set $\mathcal{V}_m^u$ are detected by the model trained under the supervision of the labeled node set $\mathcal{V}_m^l$. Each client aims to learn an anomaly scoring function $f_m(\cdot)$ that identifies whether a node is abnormal.

## 4. Methodology

This section details the proposed MV-FGAD framework for efficient and effective federated GAD. The overall framework of MV-FGAD is shown in Fig. 4. Firstly, we introduce a federated knowledge learning module (Sec. 4.1). Next, to address anomaly patterns of varying degrees, we design a multi-view learning module (Sec. 4.2). Finally, MHD-based anomaly scoring module (Sec. 4.3) is employed to quantify the deviation of each node based on the representations learned from different views.

### 4.1. Federated Knowledge Learning

We utilize GCN for feature learning and FedAvg for federated aggregation. Although more advanced feature learning methods and aggregation strategies may potentially achieve better detection performance, this work focuses on exploring an efficient and effective federated GAD framework with the simplified and general design.

**Federated Collaborative Training.** We perform federated GAD based on FedAvg (McMahan et al., 2017). Its general

form at the $i$-th client and with learning rate $\eta$ is defined as:

$$\begin{aligned}
\mathbf{W}_{t,i} &= \widetilde{\mathbf{W}}_{t,i} - \eta \nabla f\left( \widetilde{\mathbf{W}}_{t,i}, (\mathbf{A}_i, \mathbf{X}_i, \mathbf{Y}_i) \right) \\
&= \widetilde{\mathbf{W}}_{t,i} - \eta \nabla \mathcal{L}_{\text{GAD}},
\end{aligned} \qquad (2)$$

where $\eta$ denotes the learning rate, $\nabla f(\cdot)$ denotes the gradients. $\mathcal{L}_{\text{GAD}}$ is the loss in federated GAD task. $\mathbf{W}_{t,i}$ and $\widetilde{\mathbf{W}}_{t,i}$ represent the $i$-th local model and the aggregated global model received from the server in the $t$-th round, respectively. The local model of each client is updated at each communication round as follows:

$$\mathbf{W}_{t,i} \leftarrow \widetilde{\mathbf{W}}_{t,i} - \eta \nabla \mathcal{L}_{\text{GAD}}. \qquad (3)$$

The server aggregates the global model and propagates it to the participating clients for the next round:

$$\widetilde{\mathbf{W}}_{t+1} \leftarrow \sum_{i=1}^{M} \frac{n_i}{n} \mathbf{W}_{t,i}, \qquad (4)$$

where $n_i$ is the number of nodes in client $C_i$, and $n = \sum_{i=1}^{M} n_i$ represents the total number of nodes across all clients.

**Federated Knowledge Transfer.** This section describes how the global model from the final aggregation on the server is transformed into transferable knowledge for subsequent multi-view learning. Based on the federated collaborative training, $\widetilde{\mathbf{W}}_{T+1}$ is the global model parameter obtained after $T$ rounds communication through Eq. (2) - Eq. (4), from which node representation is obtained:

$$\hat{\mathbf{P}}_i = f\left( \mathbf{X}_i, \mathbf{A}_i, \widetilde{\mathbf{W}}_{T+1} \right), \qquad (5)$$

where $f(\cdot)$ is set to GCN described in Eq. (1), and it can be replaced with other types of GNNs models. To improve the effectiveness of local propagation, the final probability propagation matrix of the $i$-th client consists of the original local subgraph structure and the federated knowledge-guided structure:

$$\mathbf{P}_i = \alpha \mathbf{A}_i + (1 - \alpha)\, \hat{\mathbf{P}}_i \hat{\mathbf{P}}_i^{\top}, \qquad (6)$$

where $\alpha$ is the update hyperparameter. To alleviate the bias caused by dense propagation, we apply the Laplacian-based normalization to scale the aggregated messages:

$$\widetilde{\mathbf{P}}_i = \widetilde{\mathbf{D}}^{-\frac{1}{2}} (\mathbf{P}_i + \mathbf{I}) \widetilde{\mathbf{D}}^{-\frac{1}{2}}, \qquad (7)$$

where $(\mathbf{P}_i + \mathbf{I})$ denotes the propagation matrix with self-loop, $\widetilde{\mathbf{D}}$ is the corresponding degree matrix.

### 4.2. Multi-view Learning

**Discussion.** Based on the severity of attribute and structural anomalies, anomaly patterns in GAD can be categorized

into three aspects: 1) *Strong attribute anomalies*: arising from prominent attribute irregularities. 2) *Strong structural anomalies*: arising from significant structural irregularities. 3) *Weak anomalies*: arising from subtle attribute or structural anomalies. To effectively address the detection challenges posed by these diverse anomaly patterns, we design the multi-view learning mechanism to capture them.

**Attribute-aware Anomaly Learning.** This view is designed to target strong attribute anomalies. Due to the message-passing mechanism of GNNs described in Sec. 3, node representations are influenced by neighborhood aggregation. As a result, attribute anomalies are often obscured by dominant neighborhood patterns, making them difficult to identify. To alleviate this issue, this view learns node representations solely from attributes without incorporating neighborhood information:

$$\mathbf{H}_a = \mathcal{F}(\mathbf{X}; \boldsymbol{\Theta}_a), \tag{8}$$

where $\mathcal{F}$ is a simple MLP parameterized by $\boldsymbol{\Theta}_a$.

**Structure-aware Anomaly Learning.** This view is designed to target strong structural anomalies. Prior studies show that appropriate higher-order topological information can enhance representation learning (Tian & Zafarani, 2024; Kim et al., 2024; Wang et al., 2025a), which facilitates more effective capture of structural anomalies. In federated knowledge learning (Sec. 4.1), the optimized probabilistic propagation matrix $\widetilde{\mathbf{P}}$ implicitly encodes higher-order correlations for each node. Based on this, we perform joint knowledge-guided smoothing to obtain representations:

$$\widetilde{\mathbf{X}}^{(k)} = \left(\widetilde{\mathbf{D}}^{-\frac{1}{2}} \widetilde{\mathbf{P}} \widetilde{\mathbf{D}}^{-\frac{1}{2}}\right)^k \mathbf{X}. \tag{9}$$

Then, we obtain concatenated propagation features $\widetilde{\mathbf{X}}_{con} = \left[\widetilde{\mathbf{X}}^{(1)}, \| \dots \|, \widetilde{\mathbf{X}}^{(k)}\right]$ by concatenating the above results. Finally, we use a MLP parameterized by $\boldsymbol{\Theta}_s$ to learn the local structure representation:

$$\mathbf{H}_s = \mathcal{F}(\widetilde{\mathbf{X}}_{con}; \boldsymbol{\Theta}_s). \tag{10}$$

**Weak Anomaly-aware Dynamic Learning.** As shown in Fig. 5a, we examine the anomaly score distributions on the 6th client of the Amazon-all dataset learned by attribute-aware anomaly learning and structure-aware anomaly learning. We find the substantial overlap between normal and abnormal nodes, indicating that weak attribute or structural anomalies are difficult to capture effectively due to their subtle anomalous signals. Therefore, this view is designed to reduce such overlap by mining weak anomalies.

To balance effectiveness and efficiency, we design a learnable mechanism to dynamically model global message passing, inspired by prior message modeling strategies (Luan et al., 2022; Li et al., 2022; 2024; Liang et al., 2024b).

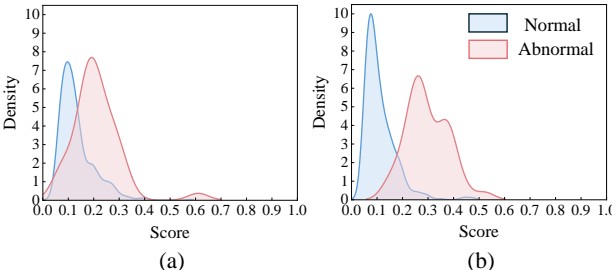

*Figure 5.* Anomaly score distribution for the 6th client on Amazon-all dataset. (a) Results without mining weak anomalies. (b) Results with mining weak anomalies.

As shown in Fig. 5b (with detailed results and analysis in Appendix E.5.1), this mechanism effectively reduces the overlap between normal and anomalous nodes while sharpening density peaks. Specifically, capturing weak anomalies requires an appropriate mechanism to amplify their gradual accumulation across layers. To this end, we adopt a $l$-layer MLP parameterized by $\boldsymbol{\Theta}_w$ to model message propagation:

$$\mathbf{H}_w^{(l)} = \mathcal{F}(\mathbf{H}_w^{(l-1)}; \boldsymbol{\Theta}_w), \tag{11}$$

where $\mathbf{H}_w^{(0)} = \mathbf{H}_s$. Based on this, we model the relationship matrix between nodes in the representation space:

$$\mathbf{R}^{(l)} = \mathbf{H}_w^{(l)}(\mathbf{H}_w^{(l)})^\top, \tag{12}$$

where $\mathbf{R}_{i,j}^{(l)}$ represents the pairwise relational similarity between node $i$ and $j$ at the $l$-th layer MLP. In general, normal nodes tend to maintain highly consistent relational patterns with their neighbors, whereas abnormal nodes are more likely to deviate from relational consistency. It is worth noting that if this signal can be effectively accumulated during training, the weak signals of weak anomalies will no longer be weak and can be captured more efficiently. To efficiently capture such characteristics, we dynamically obtain the representation signal $\widetilde{\mathbf{S}}^{(l)}$ based on the relationship similarity information $\mathbf{R}^{(l)}$:

$$\widetilde{\mathbf{S}}^{(l)} = \beta \widetilde{\mathbf{S}}^{(l-1)} + (1 - \beta)\mathbf{R}^{(l)}, \tag{13}$$

where $\widetilde{\mathbf{S}}^{(0)} = \widetilde{\mathbf{P}}$, and propagation hyperparameter $\beta \in [0, 1]$ controls the balance between $\widetilde{\mathbf{S}}^{(l-1)}$ and $\mathbf{R}^{(l)}$. To explicitly disentangle different types of relational signals, we further obtain consistency signal $\mathbf{H}_c^{(l)}$ and deviation signal $\mathbf{H}_d^{(l)}$ from the relationship matrix:

$$\mathbf{H}_c^{(l)} = \sigma(\widetilde{\mathbf{S}}^{(l)})\mathbf{H}_w^{(l)}, \quad \mathbf{H}_d^{(l)} = \sigma(-\widetilde{\mathbf{S}}^{(l)})\mathbf{H}_w^{(l)}, \tag{14}$$

where $\sigma$ is nonlinear activation function (e.g., PReLU). $\mathbf{H}_c^{(l)}$ captures coherent and stable relations, while $\mathbf{H}_d^{(l)}$ highlights relational discrepancies indicative of anomalous patterns. Then, the two complementary signals are fused with $\mathbf{H}_w^{(l+1)}$ to update the weak anomaly-aware dynamic representation:

$$\mathbf{H}_w^{(l+1)} = \mathbf{H}_w^{(l)} + \mathbf{H}_c^{(l)} + \mathbf{H}_d^{(l)}. \tag{15}$$

## 4.3. MHD-based Anomaly Scoring

Based on the multi-view learned, and inspired by the strong performance of MHD in anomaly detection (Kamoi & Kobayashi, 2020), we use it to calculate the anomaly score. For attribute representation $\mathbf{H}_a$, the MHD of node $v_i$ to surrounding nodes can be formalized as:

$$d_{a,i} = \sqrt{(\mathbf{h}_{a,i} - \boldsymbol{\mu}_a)^\top (\boldsymbol{\Sigma}_a)^{-1} (\mathbf{h}_{a,i} - \boldsymbol{\mu}_a)}, \qquad (16)$$

where $d_{a,i}$ represents the attribute representation vector of node $v_i$, $\boldsymbol{\mu}_a$ is the mean vector in the attribute representation space, and $\boldsymbol{\Sigma}_a$ is the covariance matrix of $\mathbf{H}_a$. Next, we obtain the anomaly score $s_{a,i} = \mathcal{M}(d_{a,i}) \in [0,1]$, where $\mathcal{M}(\cdot)$ is the min-max normalization function. Similarly, we obtain the anomaly scores $s_{s,i}$ and $s_{w,i}$ for node $v_i$ in the structure representation $\mathbf{H}_s$ and dynamic representation $\mathbf{H}_w$, respectively. Finally, the overall anomaly score of node $v_i$ is computed as the average of the three scores:

$$s_i = \frac{1}{3}\left(s_{a,i} + s_{s,i} + s_{w,i}\right). \qquad (17)$$

**Model Training.** For federated knowledge learning, the local training of each client is constrained by the representations $\mathbf{H}$ processed by the GCN. Eq. (16) defines the measure of deviation of node representations. Accordingly, we compute the average deviation for the normal node set $\mathcal{V}_l^n$ and abnormal node set $\mathcal{V}_l^a$ in the training data:

$$\bar{d}_{\mathcal{V}_l} = \frac{1}{|\mathcal{V}_l|} \sum_{i \in \mathcal{V}_l} d_i, \qquad (18)$$

where $\mathcal{V}_l$ is instantiated as $\mathcal{V}_l^n$ and $\mathcal{V}_l^a$, respectively, yielding $\bar{d}_{\mathcal{V}_l^n}$ and $\bar{d}_{\mathcal{V}_l^a}$. Thus, we make the distribution of normal nodes more concentrated, i.e., $\mathcal{L}_{nor} = \bar{d}_{\mathcal{V}_l^n}$. Meanwhile, to ensure that the degree of deviation of abnormal nodes is at least greater than that of normal nodes, we define $\mathcal{L}_{ano}$:

$$\mathcal{L}_{ano} = \max\left(0, \ m - \left(\bar{d}_{\mathcal{V}_l^a} - \bar{d}_{\mathcal{V}_l^n}\right)\right), \qquad (19)$$

where $m$ denotes a margin hyperparameter that enforces a minimum separation between the average deviations of abnormal and normal nodes. Finally, the loss for the federated knowledge learning phase is:

$$\mathcal{L}_f = \mathcal{L}_{nor} + \mathcal{L}_{ano}. \qquad (20)$$

For the multi-view learning phase, we calculate $\mathcal{L}_a$, $\mathcal{L}_s$ and $\mathcal{L}_w$ for the three views via Eq. (20), and combine them into the final loss:

$$\mathcal{L} = \mathcal{L}_a + \mathcal{L}_s + \mathcal{L}_w. \qquad (21)$$

Detailed algorithmic description of MV-FGAD are provided in Appendix A.1.

## 5. Experiments

### 5.1. Experimental Setup

**Datasets.** We evaluate MV-FGAD on seven real-world GAD benchmark datasets covering diverse scales and domains, including five commonly used datasets Reddit (Kumar et al., 2019; Liu et al., 2022), Tolokers (Platonov et al., 2023), Amazon (Dou et al., 2020), Amazon-all (Dou et al., 2020) and YelpChi (Rayana & Akoglu, 2015), as well as two large-scale datasets Questions (Platonov et al., 2023) and Elliptic (Weber et al., 2019; Tang et al., 2023). More details are provided in Appendix B.

**Baselines.** We compare MV-FGAD with all existing publicly available and reproducible federated GAD models, including 1) LG-FGAD (Cai et al., 2024a), 2) FGAD (Cai et al., 2024b) and 3) FedCLGN (Wu et al., 2025). Due to the limited reproducible federated GAD methods, we incorporate classic FL classification models, including 1) FedAvg (McMahan et al., 2017) 2) FedProx (Li et al., 2020), as well as representative FGL classification models, including 1) GCFL (Xie et al., 2021) and 2) FedAux (Zhuo et al., 2025). More details are provided in Appendix C.

**Evaluation and Implementations.** Following (Tang et al., 2023; Cai et al., 2024a; Qiao et al., 2025b), we use AUROC and AUPRC as evaluation metrics for GAD, and report the average AUROC/AUPRC and their standard deviations from 3 trials. All datasets are split into 20% for training, 40% for validation, and 40% for testing. To ensure a fair comparison, the same settings were applied to all baselines. More details are provided in Appendix D.

### 5.2. Effectiveness Analysis

**Superiority Results.** Table 1 compares the AUROC and AUPRC performance of MV-FGAD with baselines. We observe that 1) MV-FGAD consistently outperforms FL-based (FedAvg and FedProx) and FGL-based (GCFL+ and FedAux) classification models on most datasets. In particular, on the Amazon-all, MV-FGAD achieves improvements of at least 41.2% in AUROC and 177.8% in AUPRC, highlighting the complexity of anomaly patterns in federated GAD and the difficulty of effective anomaly mining. 2) Compared with federated GAD methods (LG-FGAD, FGAD and FedCLGN), MV-FGAD yields substantial gains on Amazon-all, exceeding the best-competing baseline LG-FGAD by 18.3% in AUROC and 96.7% in AUPRC. This indicates that neighborhood aggregation is insufficient for capturing complex anomaly patterns and highlights the necessity of multi-view learning. Although LG-FGAD achieves higher AUPRC on YelpChi, its performance is unstable due to large standard deviation. 3) Overall, MV-FGAD attains the best or second-best performance on all datasets, and achieves the strongest overall ranking.

*Table 1.* GAD performance in terms of AUROC and AUPRC (in percent, mean±std). Highlighted are the results ranked **first** and second, and **large std** is emphasized. "OOM" indicates out-of-memory on a 24GB GPU. "Rank" indicates the average ranking over five datasets (performance with large std are assigned the lowest rank).

| Metric | Model | Reddit | Tolokers | Amazon | Amazon-all | YelpChi | Rank |
|---|---|---|---|---|---|---|---|
| AUROC | FedAvg (McMahan et al., 2017) | 51.12±0.63 | 37.44±0.29 | 38.51±1.77 | 50.86±0.04 | 51.72±0.42 | 6.8 |
| | FedProx (Li et al., 2020) | 51.72±1.99 | 37.42±0.16 | 42.41±1.08 | 50.76±0.09 | 50.91±0.04 | 7.2 |
| | GCFL+ (Xie et al., 2021) | 50.90±0.41 | 40.21±2.51 | 43.43±1.41 | 53.98±4.71 | 51.67±0.30 | 6.4 |
| | FedAux (Zhuo et al., 2025) | 53.50±6.44 | 53.59±1.11 | 62.67±0.65 | 64.37±1.42 | 53.70±0.05 | 2.8 |
| | LG-FGAD (Cai et al., 2024a) | 57.35±0.52 | 53.53±4.54 | 57.74±4.62 | 76.83±0.17 | 50.50±0.71 | 3.6 |
| | FGAD (Cai et al., 2024b) | 52.12±1.19 | 51.53±0.70 | 55.97±0.17 | 64.67±0.73 | 52.17±0.34 | 4.0 |
| | FedCLGN (Wu et al., 2025) | 53.93±1.02 | 51.36±0.24 | 53.04±0.89 | 56.10±0.06 | 52.53±0.44 | 4.2 |
| | MV-FGAD (ours) | **59.27±1.03** | **55.73±1.01** | **78.33±0.74** | **90.92±0.77** | **64.25±0.28** | **1.0** |
| AUPRC | FedAvg (McMahan et al., 2017) | 3.81±0.07 | 18.23±0.17 | 5.82±0.15 | 7.40±0.62 | 16.11±0.36 | 6.4 |
| | FedProx (Li et al., 2020) | 3.88±0.11 | 18.05±0.14 | 6.25±0.25 | 5.82±1.52 | 15.71±0.11 | 6.6 |
| | GCFL+ (Xie et al., 2021) | 3.83±0.06 | 19.39±1.08 | 6.55±0.22 | 7.03±3.89 | 16.10±0.16 | 6.0 |
| | FedAux (Zhuo et al., 2025) | 1.81±0.25 | 22.99±1.04 | 10.60±0.47 | 14.59±0.61 | 17.51±0.56 | 4.2 |
| | LG-FGAD (Cai et al., 2024a) | **6.60±1.59** | 21.55±4.10 | 15.24±4.76 | 20.61±1.11 | 33.36±**19.82** | 3.8 |
| | FGAD (Cai et al., 2024b) | 3.85±0.09 | 23.32±0.68 | 20.86±0.64 | 10.34±0.65 | 18.04±2.11 | 2.8 |
| | FedCLGN (Wu et al., 2025) | 3.42±0.12 | 22.13±0.28 | 7.76±0.55 | 7.85±0.24 | 16.27±0.56 | 5.0 |
| | MV-FGAD (ours) | 5.54±0.51 | **23.56±0.12** | **21.91±1.63** | **40.53±1.74** | **22.60±0.43** | **1.2** |

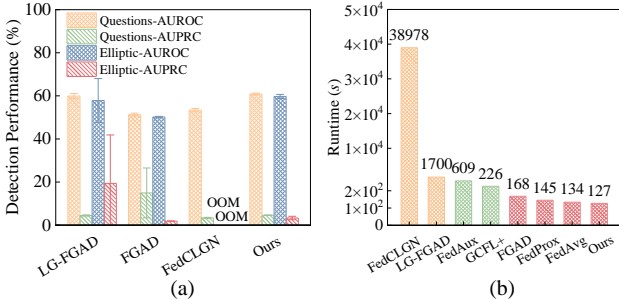

*Figure 6.* Scalability and time analysis. (a) GAD performance on large-scale datasets. (b) Runtime comparison.

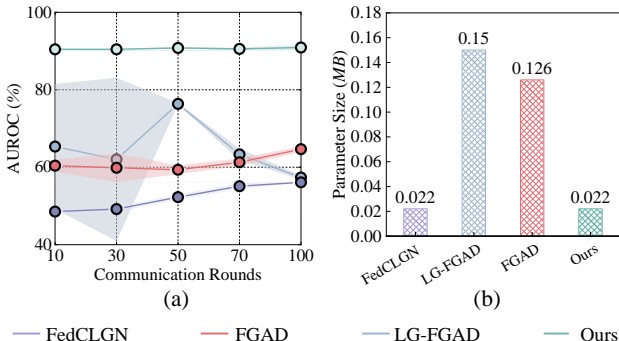

*Figure 7.* Communication efficiency analysis. (a) AUROC performance vs. communication rounds. (b) Parameter upload/download size.

**Scalability Analysis.** In Fig. 6a, we observe the scalability of existing GAD models with MV-FGAD on the large-

*Table 2.* Ablation study for multi-view learning. Highlighted are the results ranked **first** and second.

| Metric | AAL | SAL | WADL | Reddit | Tolokers | Amazon |
|---|---|---|---|---|---|---|
| AUROC | ✓ | | | 55.47±0.58 | 50.16±0.78 | 75.62±0.38 |
| | | ✓ | | 56.81±0.23 | 51.35±2.03 | 64.76±0.58 |
| | | | ✓ | 57.08±2.91 | 52.09±1.58 | 72.43±1.25 |
| | ✓ | ✓ | | 57.79±1.36 | 52.20±0.94 | 76.75±0.56 |
| | | ✓ | ✓ | 58.14±1.52 | 53.29±1.42 | 77.06±1.01 |
| | ✓ | | ✓ | 58.15±1.47 | 52.28±0.31 | 68.77±2.10 |
| | ✓ | ✓ | ✓ | **59.27±1.03** | **55.73±1.01** | **78.33±0.74** |
| AUPRC | ✓ | | | 5.05±0.66 | 21.57±0.31 | 18.56±0.71 |
| | | ✓ | | 5.18±0.08 | 20.05±0.56 | 16.04±0.21 |
| | | | ✓ | 5.14±0.30 | 20.24±1.12 | 17.29±2.12 |
| | ✓ | ✓ | | 5.23±0.25 | 22.23±0.27 | 20.84±1.57 |
| | | ✓ | ✓ | 5.34±0.54 | 22.40±0.88 | 21.50±1.40 |
| | ✓ | | ✓ | 5.37±0.05 | 22.42±0.34 | 17.44±1.46 |
| | ✓ | ✓ | ✓ | **5.54±0.51** | **23.56±0.12** | **21.91±1.63** |

*Table 3.* Ablation study for anomaly scoring. Highlighted are the results ranked **first** and second.

| Metric | Method | Reddit | Tolokers | Amazon |
|---|---|---|---|---|
| AUROC | Euclidean Distance | 56.79±1.49 | 54.67±1.46 | 71.11±0.99 |
| | Cosine Similarity | 55.66±1.43 | 55.20±0.87 | 72.19±0.18 |
| | MHD | **59.27±1.03** | **55.73±1.01** | **78.33±0.74** |
| AUPRC | Euclidean Distance | 5.02±1.44 | 23.17±1.24 | 20.83±1.33 |
| | Cosine Similarity | 5.16±0.21 | 23.31±0.41 | 20.85±1.38 |
| | MHD | **5.54±0.51** | **23.56±0.12** | **21.91±1.63** |

scale Questions and Elliptic datasets. We find that both LG-FGAD and FGAD suffer from substantial performance fluctuations, while FedCLGN runs out-of-memory (OOM) on Elliptic. In contrast, MV-FGAD consistently demon-

strates efficient and effective detection performance.

**Ablation Study.** To evaluate the contributions of Attribute-aware Anomaly Learning (**AAL**), Structure-aware Anomaly Learning (**SAL**), and Weak Anomaly-aware Dynamic Learning (**WADL**), we modify MV-FGAD by retaining selected views. As shown in Table 2 (full results in Appendix E.1.1), no single view fully captures all anomaly patterns, while combining any two views consistently improves performance. Notably, WADL achieves consistent performance improvements across datasets with varying levels of weak anomalies. Furthermore, we evaluate the MHD method for GAD measurement by comparing it with cosine similarity (Qiao & Pang, 2023) and graph-specific Euclidean distance, as shown in Table 3 (full results in Appendix E.1.2). While all three metrics show potential, MHD achieves a better balance between AUROC and AUPRC, demonstrating superior effectiveness across datasets. In addition, the Appendix E.1.3 investigates different multi-view aggregation strategies, showing that mean aggregation better preserves subtle anomaly signals and improves model stability.

### 5.3. Efficiency Analysis

**Complexity Analysis.** The time complexity analysis of MV-FGAD is provided in Appendix A.2.

**Time Analysis.** In Fig. 6b, we compare the time of all models on the Questions dataset. We observe that MV-FGAD achieves a significant advantage in runtime efficiency among all compared models. In particular, its runtime accounts for only 0.33% of that of FedCLGN, demonstrating the high efficiency of our proposed framework. Moreover, as shown in Fig. 6a. FedCLGN even encounter OOM issues on the larger-scale Elliptic dataset.

**Communication Efficiency Analysis.** In Fig. 7, we analyze the communication efficiency of MV-FGAD with federated GAD baselines on the Amazon-all. MV-FGAD achieves strong AUROC within only 10 communication rounds (AUPRC results in Appendix E.2) and maintains stable performance as rounds increase, consistently outperforming competing methods. Moreover, it requires significantly fewer upload/download parameters, comparable to FedCLGN and less than one-fifth of FGAD and LG-FGAD.

### 5.4. Other Analysis

**Sensitivity Analysis.** We analyze the impact of the update hyperparameter $\alpha$ and the propagation hyperparameter $\beta$ on model performance. Fig. 8 presents AUROC results under different $\alpha$ and $\beta$, with AUPRC results provided in Appendix E.3. We observe that 1) when $\alpha = 0.5$, the original graph structure provides a stable propagation basis while federated knowledge guidance ensures both efficient structure-aware anomaly learning and effective

weak anomaly-aware dynamic learning. 2) Similarly, when $\beta = 0.5$, normal nodes preserve consistent relational patterns, whereas abnormal nodes exhibit unstable signals that hinder the accumulation of stable relational information.

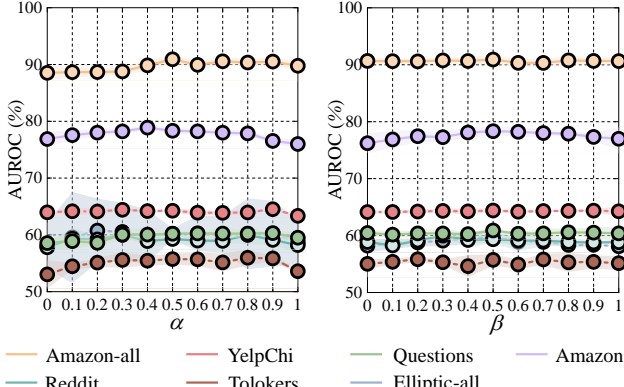

*Figure 8.* Analysis of hyperparameter $\alpha$ and $\beta$.

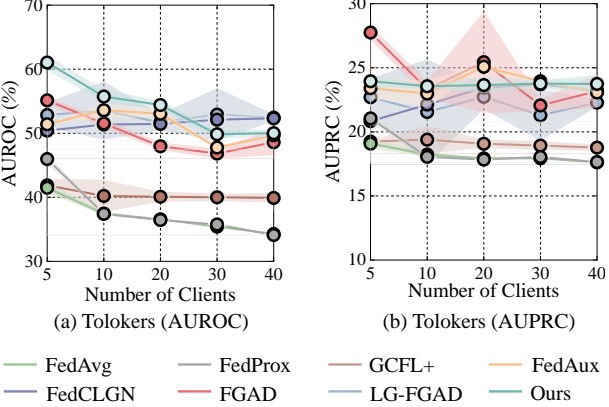

*Figure 9.* Analysis of the number of clients.

**Robustness Analysis with Multiple Runs.** Table 1 and Fig. 6a show the standard deviations of AUROC and AUPRC over three runs. Although LG-FGAD and FGAD achieve strong performance on some datasets, they exhibit substantial performance variance. For example, LG-FGAD attains an AUPRC standard deviation of 22.47 on Elliptic, indicating limited efficiency. Therefore, such results are assigned the lowest rank. In contrast, MV-FGAD consistently maintains low variance across all datasets, demonstrating superior robustness and effectiveness.

**Performance with Different Client Number.** We evaluate MV-FGAD across different numbers of clients on the Tolokers dataset, as shown in Fig. 9, and results on YelpChi are provided in Appendix E.4. As the number of clients increases, MV-FGAD shows only marginal performance degradation and consistently outperforms other methods in both effectiveness and robustness.

**Case Study for Multi-view Learning.** We examine the

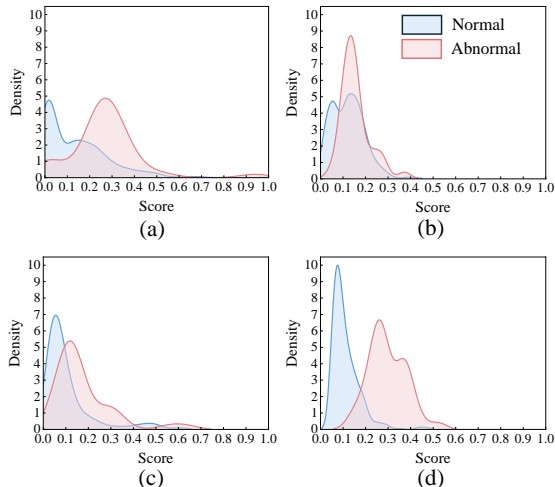

*Figure 10.* Case study for multi-view learning. (a) Attribute-aware anomaly learning. (b) Structure-aware anomaly learning. (c) Weak anomaly-aware dynamic learning. (d) Multi-view learning.

anomaly score distributions from different views on the Amazon-all dataset. Results for the 6th client are shown in Fig. 10, with detailed results in Appendix E.5.2. Multi-view learning integrates the strengths of each view, yielding higher peaks and less overlap between normal and anomalous nodes distributions.

## 6. Conclusion

This paper investigates the unreliability of existing federated GAD methods that assume neighborhood aggregation sufficiently captures anomalous information, and pioneers the identification of detection bottlenecks caused by weak anomalies. To address these issues, we revisit federated GAD and propose MV-FGAD, an efficient and effective federated GAD framework. We present the first systematic analysis of complex anomaly patterns across data silos, including strong attribute anomalies, strong structural anomalies and weak anomalies. To capture them, we design a multi-view learning framework and adopt the MHD metric to measure node deviations in each view, aggregating multiple views to obtain final anomaly scores. Extensive experiments across multiple datasets demonstrate the effectiveness and efficiency of MV-FGAD. However, label scarcity in real-world scenarios further complicates federated GAD training. In future work, we will explore efficient and effective federated GAD in fully unsupervised learning.

## Acknowledgement

This work is supported by the National Science Fund for Distinguished Young Scholars of China (No. 62325604), the National Natural Science Foundation of China (No. 62441618, 62506371, 62276271, 62572480), and the Major Program Project of Xiangjiang Laboratory (No. 24XJJ-CYJ01002).

## Impact Statement

This paper presents work whose goal is to advance the field of Machine Learning. There are many potential societal consequences of our work, none which we feel must be specifically highlighted here.

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

# A. Algorithm and Complexity

## A.1. Algorithmic Description

The detailed algorithm of MV-FGAD is described in Algo. 1 and Algo. 2.

---

**Algorithm 1** Federated Knowledge Learning

---

 1: */* Federated Collaborative Training */*
 2: **for** each communication round $t = 1, \ldots, T$ **do**
 3:     **for parallel** each client $m = 1, \ldots, M$ **do**
 4:         **for** training epoch $e = 1, \ldots, E$ **do**
 5:             Update local model weight via Eq. (2);
 6:         **end for**
 7:     **end for**
 8:     Server-side model aggregation via Eq. (3), (4);
 9: **end for**
10: Broadcast the global model to clients;
11: */* Federated Knowledge Transfer */*
12: **for parallel** each client $m = 1, \ldots, M$ **do**
13:     Calculate the probability propagation matrix $\tilde{\mathbf{P}}_i$ via Eq. (5), (6), (7);
14:     Calculate loss $\mathcal{L}_f$ via Eq. (20);
15:     Update client parameters via gradient descent.
16: **end for**

---

**Algorithm 2** Multi-view learning & MHD-based Anomaly Scoring

---

 1: **for parallel** each client $m = 1, \ldots, M$ **do**
 2:     **for** training epoch $e = 1, \ldots, E$ **do**
 3:         Execute attribute-aware anomaly learning to obtain $\mathbf{H}_a$ via Eq. (8);
 4:         Execute structure-aware anomaly learning to obtain $\mathbf{H}_s$ via Eq. (10);
 5:         Execute weak anomaly-aware dynamic learning to obtain $\mathbf{H}_w$ via Eq. (15);
 6:         Calculate the anomaly score $s_i$ via Eq. (17);
 7:     **end for**
 8:     Calculate loss $\mathcal{L}$ via Eq. (21);
 9:     Update client parameters via gradient descent.
10: **end for**

---

## A.2. Complexity Analysis

The overall complexity of MV-FGAD consists of three components: federated knowledge learning, multi-view learning and MHD-based anomaly scoring. For federated knowledge learning, the model is trained over $T$ communication rounds. Since clients perform local training in parallel, the overall complexity is $\mathcal{O}(T(\bar{\mathcal{E}}d' + \bar{n}d'^2))$, where the first term is used for message passing, and the second term is used for feature mapping. Here, $\bar{\mathcal{E}}$ and $\bar{n}$ denote the average numbers of edges and nodes across all clients, respectively, and $d'$ is the representation dimension. For multi-view learning, the main time complexity stems from weak anomaly-aware dynamic learning, whose core complexity lies in relation matrix modeling, taking $\mathcal{O}(\bar{n}^2d' + \bar{n}d'^2)$, where the first term is used for relation matrix modeling and fusion, and the second term is used for updating the linear layer. For MHD-based anomaly scoring, the computational complexity scales linearly with respect to the number of reference nodes, i.e., $\mathcal{O}(\bar{n}d'^2)$. Therefore, the overall time complexity of MV-FGAD is $\mathcal{O}(T(\bar{\mathcal{E}}d' + \bar{n}d'^2) + \bar{n}^2d' + \bar{n}d'^2 + \bar{n}d'^2) \rightarrow \mathcal{O}(T(\bar{\mathcal{E}}d' + \bar{n}d'^2) + \bar{n}^2d')$.

# B. Detailed Description of Datasets

The statistical details of the datasets used in our experiments are summarized in Table 4. Detailed descriptions of each dataset are as follows:

*Table 4.* The statistics of the datasets. Misc. denotes that node features are composed of heterogeneous attributes, such as categorical, numerical, and temporal information.

| Datasets | # Nodes | # Edges | # Attributes | # Anomaly (%) | Relation Type | Feature Type |
|---|---|---|---|---|---|---|
| Reddit | 10,984 | 168,016 | 64 | 3.3% | Social Media | Text Embedding |
| Tolokers | 11,758 | 519,000 | 10 | 21.8% | Work Collaboration | Misc. Information |
| Amazon | 10,244 | 175,608 | 25 | 6.7% | Co-review | Misc. Information |
| Amazon-all | 11,944 | 4,398,392 | 25 | 9.5% | Co-review | Misc. Information |
| YelpChi | 45,954 | 3,846,979 | 32 | 14.5% | Co-review | Misc. Information |
| Questions | 48,921 | 153,540 | 301 | 3.0% | Question Answering | Text Embedding |
| Elliptic | 203,769 | 234,355 | 166 | 2.2% | Bitcoin Transaction | Misc. Information |

- **Reddit** (Kumar et al., 2019; Liu et al., 2022) records user posts across different subreddits over a one-month period. This dataset is constructed as a user–subreddit interaction graph and includes verified labels indicating banned users. It focuses on the 1,000 most active subreddits and the 10,000 most engaged users, together with all interaction records between them.

- **Tolokers** (Platonov et al., 2023) is collected from the Toloka crowd-sourcing platform. Nodes represent workers who have participated in at least one of the 13 selected projects, and node features are constructed from workers' profile information and task performance statistics. An edge connects two workers if they have participated in the same task.

- **Amazon** (Dou et al., 2020) aims to identify paid users who post fraudulent reviews for musical instrument products on Amazon.com. Users whose reviews receive more than 80% helpful votes are labeled as normal users, while those with less than 20% helpful votes are regarded as fraudulent users. This graph contains three types of relations: U–P–U (users who have reviewed at least one common product), U–S–U (users who have given at least one identical star rating within a one-week period), and U–V–U (users whose review similarity ranks in the top 5%). In this paper, Amazon refers to the U-P-U relationship type.

- **Amazon-all** (Dou et al., 2020) treats all connections in U–P–U (users who have reviewed at least one common product), U–S–U (users who have given at least one identical star rating within a one-week period), and U–V–U (users whose review similarity ranks in the top 5%) as the same type of edge and utilize all available links, whereas Amazon only considers the U–P–U relationship.

- **YelpChi** (Rayana & Akoglu, 2015) aims to identify anomalous reviews that unfairly promote or defame products or businesses on Yelp.com. This graph contains three types of edges: R-U-R (reviews posted by the same user), R–S–R (reviews for the same product with identical star ratings), and R–T–R (reviews for the same product posted within the same month). In this paper, we focus on the R–U–R relation.

- **Questions** (Platonov et al., 2023) is collected from the question-and-answer website Yandex Q and aims to predict which users remain active on the platform at the end of a one-year period (from September 2021 to August 2022). Nodes represent users, and node features are obtained by averaging the FastText embeddings of words in user descriptions. An edge connects two users if they have question–answer interactions during this period. This dataset focuses on users who are interested in the "Medicine" topic.

- **Elliptic** (Weber et al., 2019; Tang et al., 2023) maps Bitcoin transactions to real-world entities associated with legitimate categories, such as exchanges, wallet providers, miners, and legal service organizations, as well as entities associated with illicit categories, including scams, malware, terrorist organizations, ransomware, and Ponzi schemes. The dataset consists of a graph in which nodes represent transactions and edges denote the flow of Bitcoin funds. It contains over 200,000 Bitcoin transactions (nodes), 234,000 directed payment flows (edges), and 166-dimensional node features.

## C. Detailed Description of Baselines

We compare MV-FGAD with all three publicly available and reproducible federated GAD models. Given the limited availability of domain-specific methods, we further include two standard FL classification models and two representative

FGL classification models for comparison. Notably, to avoid performance degradation from treating GAD as a binary classification task, classification models also compute anomaly scores using the MHD.

For federated GAD methods, we compare MV-FGAD with all existing baselines that have publicly available implementations, covering both node-level and graph-level models.

- **LG-FGAD** (Cai et al., 2024a) enhances anomaly detection through a local–global anomaly awareness mechanism that leverages mutual information maximization and minimization between normal and generated anomalous graphs at both node and graph levels. To alleviate non-IID effects while preserving client personalization, it further adopts a dual knowledge distillation strategy over logits and embedding distributions, and improves communication efficiency by engaging only lightweight student models in federated collaboration.

- **FGAD** (Cai et al., 2024b) improves anomaly detection by generating self-perturbed anomalous graphs and training the detector to distinguish them from normal graphs. To alleviate non-IID effects while preserving local personalization, it adopts a self-boosted knowledge distillation strategy within a student–teacher framework, and employs an efficient federated learning mechanism to streamline local model capacity and reduce communication overhead.

- **FedCLGN** (Wu et al., 2025) aggregates negative pair representations of pseudo-labeled anomalous nodes from multiple clients into a global negative pair pool on the server and leverages graph diffusion to capture both global and local structural patterns. By enhancing the discrimination between positive and negative pairs while preserving privacy, FedCLGN achieves potential and robust anomaly detection.

For FL classification methods, we compare MV-FGAD with two existing standard models.

- **FedAvg** (McMahan et al., 2017) is a FL learning algorithm that enables multiple clients to collaboratively train a global model without sharing data. Each client performs local updates using its own data, and the server periodically averages these updates to form the global model. This method preserves data privacy, handles unbalanced and non-IID data distributions, and significantly reduces communication overhead, making it highly efficient across data silos.

- **FedProx** (Li et al., 2020) extends FedAvg by introducing a proximal term in the local objective, which regularizes client updates toward the global model. This design addresses statistical heterogeneity with non-IID data and systems heterogeneity with variable client capabilities, stabilizing convergence and improving performance in FL.

For FGL classification methods, we compare MV-FGAD with two existing representative models.

- **GCFL+** (Xie et al., 2021) improves upon GCFL by considering multi-round gradient sequences using dynamic time warping, while GCFL dynamically clusters clients based on GNN gradients to reduce structural and feature heterogeneity. These frameworks enable more homogeneous collaboration among clients, leading to improved model performance and convergence across heterogeneous graph datasets.

- **FedAux** (Zhuo et al., 2025) uses learnable auxiliary projection vector (APV) to capture client-specific information. Each client trains a local GNN alongside an APV that projects node embeddings into a 1D latent space, which is refined through soft-sorting and lightweight 1D convolution. The server aggregates client updates based on APV similarities, producing personalized models while preserving cross-client knowledge and protecting privacy.

## D. Details of Implementation

**Parameter Settings.** The experiments are conducted with 10 clients by default, and we further analyze performance under varying numbers of clients in Appendix E.4. The model is trained for 200 epochs with a weight decay of 5e-4. The learning rate is set to 5e-3 for Tolokers and Elliptic, and 1e-3 for the remaining datasets. The hidden dimension is fixed to 64, and 100 rounds of federated learning are performed for all datasets. The hyperparameters $\alpha$ and $\beta$ are both set to 0.5, and the detailed analysis provided in Appendix E.3.

**Metrics.** Following centralized GAD studies (Tang et al., 2023; Cai et al., 2024a; Qiao et al., 2025b), we employ two widely used and complementary metrics to evaluate the anomaly detection performance of federated GAD models, including the area under the receiver operating characteristic curve (AUROC) and the area under the precision-recall curve (AUPRC).

*Table 5.* Ablation study for multi-view learning. Highlighted are the results ranked **first** and second.

| Metric | AAL | SAL | WADL | Reddit | Tolokers | Amazon | Amazon-all | YelpChi | Questions | Elliptic |
|---|---|---|---|---|---|---|---|---|---|---|
| AUROC | ✓ | | | 55.47±0.58 | 50.16±0.78 | 75.62±0.38 | 86.58±0.90 | 59.74±0.14 | 56.58±0.74 | 44.17±0.32 |
| | | ✓ | | 56.81±0.23 | 51.35±2.03 | 64.76±0.58 | 79.00±0.62 | 60.63±0.98 | 59.60±0.33 | 59.40±7.94 |
| | | | ✓ | 57.08±2.91 | 52.09±1.58 | 72.43±1.25 | 85.43±0.58 | 61.03±0.51 | 59.85±0.99 | **59.96±4.53** |
| | ✓ | ✓ | | 57.79±1.36 | 52.20±0.94 | 76.75±0.56 | 90.67±0.58 | 62.80±0.65 | 60.47±0.59 | 50.74±4.79 |
| | | ✓ | ✓ | 58.14±1.52 | 53.29±1.42 | 77.06±1.01 | 82.77±0.62 | 62.84±0.61 | 60.21±0.54 | 59.71±4.60 |
| | ✓ | | ✓ | 58.15±1.47 | 52.28±0.31 | 68.77±2.10 | 90.04±0.57 | 63.17±0.19 | 60.42±0.64 | 59.95±5.60 |
| | ✓ | ✓ | ✓ | **59.27±1.03** | **55.73±1.01** | **78.33±0.74** | **90.92±0.77** | **64.25±0.28** | **60.89±0.38** | 59.72±0.95 |
| AUPRC | ✓ | | | 5.05±0.66 | 21.57±0.31 | 18.56±0.71 | 33.38±1.50 | 19.69±0.04 | 4.37±0.35 | 1.85±0.01 |
| | | ✓ | | 5.18±0.08 | 20.05±0.56 | 16.04±0.21 | 27.78±3.86 | 19.90±1.08 | 4.04±0.35 | 2.98±1.09 |
| | | | ✓ | 5.14±0.30 | 20.24±1.12 | 17.29±2.12 | 32.19±5.61 | 19.37±0.34 | 4.13±0.19 | **3.69±1.53** |
| | ✓ | ✓ | | 5.23±0.25 | 22.23±0.27 | 19.84±1.57 | 35.88±0.91 | 20.45±0.26 | 4.34±0.14 | 2.15±0.29 |
| | | ✓ | ✓ | 5.34±0.54 | 22.40±0.88 | 21.50±1.40 | 30.77±5.52 | 20.86±0.48 | 4.45±0.21 | 3.58±1.17 |
| | ✓ | | ✓ | 5.37±0.05 | 22.42±0.34 | 17.44±1.46 | 39.84±2.17 | 21.50±0.12 | 4.31±0.06 | 3.39±0.82 |
| | ✓ | ✓ | ✓ | **5.54±0.51** | **23.56±0.12** | **21.91±1.63** | **40.53±1.74** | **22.60±0.43** | **4.58±0.11** | 3.32±0.79 |

Higher AUROC and AUPRC values indicate better GAD performance. We report the mean AUROC and AUPRC over 3 trials to evaluate the efficiency and effectiveness of the models.

**Computing Infrastructures.** We implemented our method using PyTorch 2.0.1+cu118, PyTorch Geometric (PyG) 2.6.1 and DGL 1.1.2+cu118. All experiments were conducted on a workstation with an Intel Core i9-14900KF CPU and an NVIDIA GeForce RTX 3090 GPU.

# E. Additional Experimental Results

## E.1. Detailed Ablation Study

### E.1.1. ABLATION STUDY FOR MULTI-VIEW LEARNING

To evaluate the contributions of Attribute-aware Anomaly Learning (AAL), Structure-aware Anomaly Learning (SAL), and Weak Anomaly-aware Dynamic Learning (WADL), we modify MV-FGAD by retaining selected views, as shown in Table 5. We observe that 1) No single view can fully capture all anomaly patterns, confirming that the complex anomalies in federated GAD are difficult to effectively detect using single-view detection strategy. 2) Combining any two views consistently improves performance, indicating that more effective anomaly mining requires a more comprehensive consideration of diverse anomaly patterns. Notably, WADL achieves consistent detection improvements across datasets with varying levels of weak anomalies. 3) Although the WADL view achieves the best performance on the Elliptic dataset, multi-view learning that separately targets strong attribute anomalies, strong structural anomalies, and weak anomalies is more efficient and effective across all datasets.

### E.1.2. ABLATION STUDY FOR ANOMALY SCORING

We evaluate the applicability of MHD for GAD measurement by comparing it with cosine similarity (Qiao & Pang, 2023) and graph-specific Euclidean distance, as shown in Table 6. We observe that 1) All three metrics show the potential to identify anomalous nodes in the representation space, owing to their ability to quantify similarity or deviation in graph data. 2) Cosine similarity and Euclidean distance struggle to maintain a favorable balance between AUROC and AUPRC, often excelling in only one of the two metrics. 3) Across all datasets, MHD achieves a better balance between AUROC and AUPRC, demonstrating superior efficiency and effectiveness for GAD tasks.

### E.1.3. ABLATION STUDY FOR MULTI-VIEW AGGREGATION

To investigate the effectiveness of different aggregation strategies in multi-view computation, we compare max-based and mean-based aggregation in Table 7. Although max-based aggregation is a feasible design, it underperforms mean-based aggregation in capturing comprehensive anomaly information. This may be because max aggregation is more sensitive to noisy information, whereas mean aggregation better preserves subtle anomaly signals and improves model stability.

*Table 6.* Ablation study for anomaly scoring. Highlighted are the results ranked **first** and second.

| Metric | Method | Reddit | Tolokers | Amazon | Amazon-all | YelpChi | Questions | Elliptic |
|---|---|---|---|---|---|---|---|---|
| AUROC | Euclidean Distance | 56.79±1.49 | 54.67±1.46 | 71.11±0.99 | 71.05±2.00 | 60.83±0.08 | 57.22±2.82 | **71.71±2.06** |
| | Cosine Similarity | 55.66±1.43 | 55.20±0.87 | 72.19±0.18 | 71.13±2.31 | 60.84±0.41 | 58.29±3.69 | 70.05±2.86 |
| | MHD | **59.27±1.03** | **55.73±1.01** | **78.33±0.74** | **90.92±0.77** | **64.25±0.28** | **60.89±0.38** | 59.72±0.95 |
| AUPRC | Euclidean Distance | 5.02±1.44 | 23.17±1.24 | 20.83±1.33 | 19.04±3.88 | 20.85±0.49 | 6.40±0.62 | 5.53±0.49 |
| | Cosine Similarity | 5.16±0.21 | 23.31±0.41 | 20.85±1.38 | 12.69±1.43 | 20.98±0.34 | **6.42±0.62** | 5.58±0.92 |
| | MHD | **5.54±0.51** | **23.56±0.12** | **21.91±1.63** | **40.53±1.74** | **22.60±0.43** | 4.58±0.11 | 3.32±0.79 |

*Table 7.* Ablation study for multi-view computing. Highlighted are the results ranked **first**.

| Metric | Method | Reddit | Tolokers | Amazon | Amazon-all | YelpChi | Questions | Elliptic |
|---|---|---|---|---|---|---|---|---|
| AUROC | MV-FGAD (Max) | **60.28±1.08** | 51.32±0.63 | 78.17±0.11 | 89.02±1.23 | 63.49±0.88 | 59.81±0.78 | 58.88±0.69 |
| | MV-FGAD (Mean) | 59.27±1.03 | **55.73±1.01** | **78.33±0.74** | **90.92±0.77** | **64.25±0.28** | **60.89±0.38** | **59.72±0.95** |
| AUPRC | MV-FGAD (Max) | 4.98±0.15 | 21.71±0.33 | **22.38±0.34** | 33.94±0.83 | 21.67±0.89 | 4.21±0.32 | 3.23±0.24 |
| | MV-FGAD (Mean) | **5.54±0.51** | **23.56±0.12** | 21.91±1.63 | **40.53±1.74** | **22.60±0.43** | **4.58±0.11** | **3.32±0.79** |

### E.2. Detailed Communication Efficiency Analysis

We analyze the communication efficiency of MV-FGAD with federated GAD baselines on the Amazon-all dataset, as shown in Fig. 11. MV-FGAD achieves strong detection performance within only 10 communication rounds, while maintaining stable performance as the number of communication rounds increases, consistently outperforming other competing methods in anomaly detection tasks. Moreover, in terms of parameter upload/download sizes, MV-FGAD achieves efficiency comparable to FedCLGN, while its parameter scale is less than one-fifth of that of FGAD and LG-FGAD.

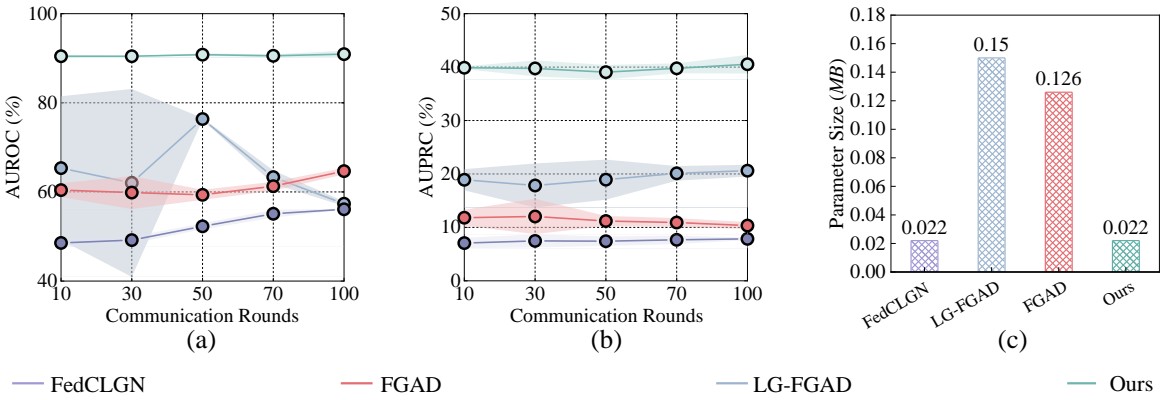

*Figure 11.* Communication efficiency analysis. (a) AUROC performance vs. communication rounds. (b) AUPRC performance vs. communication rounds. (c) Parameter upload/download size.

### E.3. Detailed Sensitivity Analysis

We analyze the impact of the update hyperparameter $\alpha$ and the propagation hyperparameter $\beta$ on the performance of MV-FGAD. The AUROC and AUPRC results under different values of $\alpha$ and $\beta$ are shown in Fig. 12. We can observe that 1) When the update hyperparameter $\alpha$ is set to 0.5, the original graph structure provides the reliable propagation foundation, while the federated knowledge–guided information ensures both the stability of structure-aware anomaly learning and the effectiveness of weak anomaly-aware dynamic learning in MV-FGAD. In contrast, the excessively large $\alpha$ causes the propagation to degenerate toward the original structural information, whereas the overly small $\alpha$ leads to excessively broad propagation range, thereby amplifying noisy signals. 2) Similarly, when the propagation hyperparameter $\beta$ is set to 0.5, normal nodes exhibit highly consistent relational patterns across successive iterations, while abnormal nodes display more volatile signals and thus fail to form stable accumulated relational information.

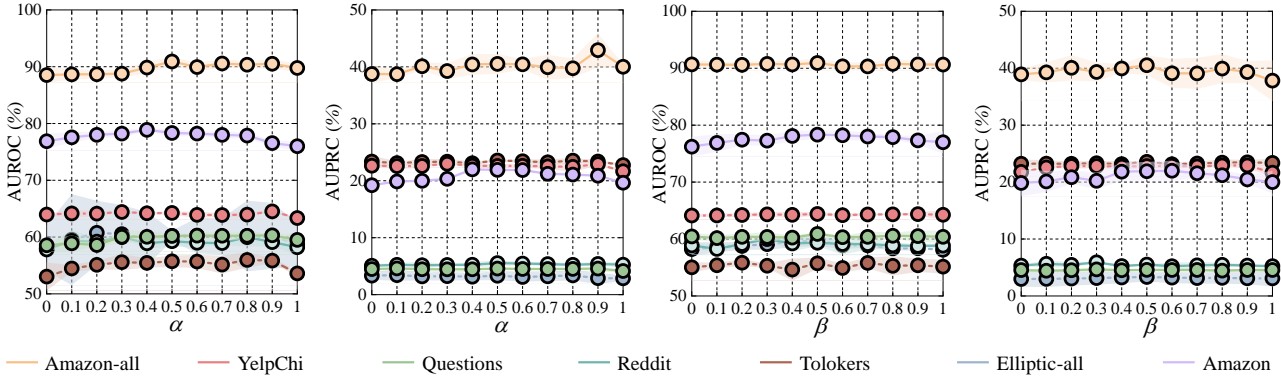

*Figure 12.* Analysis of hyperparameter $\alpha$ and $\beta$.

## E.4. Detailed Performance with Different Client Number

We evaluate the performance of MV-FGAD under varying numbers of clients on the Tolokers and YelpChi datasets, as shown in Fig. 13. We observe that 1) As the number of clients increases, all methods experience certain performance fluctuations or degradation. However, MV-FGAD is the least affected by the number of clients and consistently remains competitive. 2) Some methods (such as FGAD and LG-FGAD) exhibit large variations in performance standard deviation, resulting in reduced effectiveness, whereas MV-FGAD maintains superior effectiveness and robustness compared to baselines.

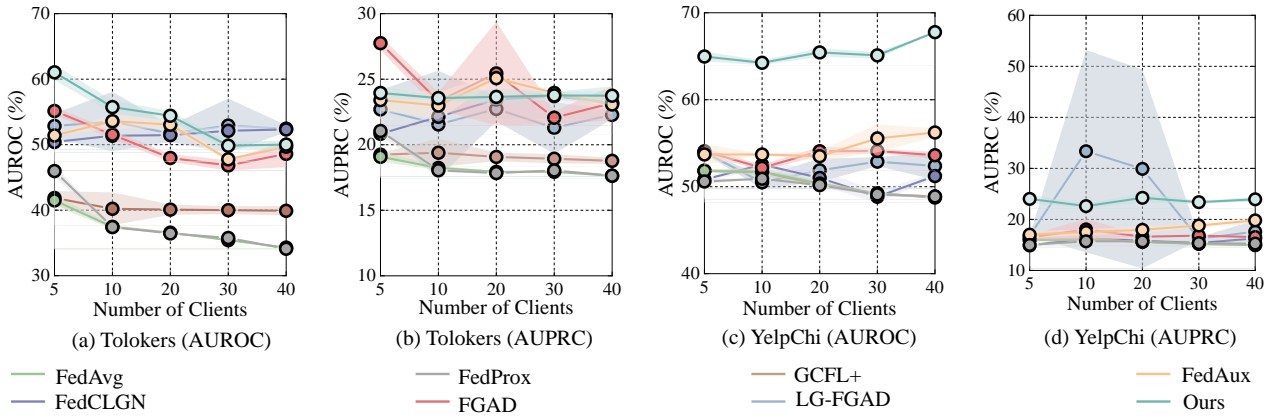

*Figure 13.* Analysis of the number of clients.

## E.5. Detailed Case Study

### E.5.1. CASE STUDY FOR WEAK ANOMALY-AWARE DYNAMIC LEARNING

On the Amazon-all dataset, we select clients with even indices to compare models that consider only strong attribute and strong structural anomalies with those that jointly model all three types of anomalies, as shown in Fig. 14. We observe that 1) The overlap between normal and anomalous distributions varies depending on the level of weak anomalies across clients, highlighting the challenge of detecting weak anomalies. However, weak anomaly-aware dynamic learning consistently enables effective mining of these weak anomalies. 2) Modeling weak anomalies increases the peak of the normal node distribution. This further expands the distribution boundary between normal and abnormal nodes.

### E.5.2. CASE STUDY FOR MULTI-VIEW LEARNING

On the Amazon-all dataset, we select clients with even indices to compare models that individually perform attribute-aware anomaly learning, structure-aware anomaly learning, and weak anomaly-aware dynamic learning with MV-FGAD, which jointly integrates all three views, as shown in Fig. 15. We observe that 1) Across different client subgraphs, the anomaly

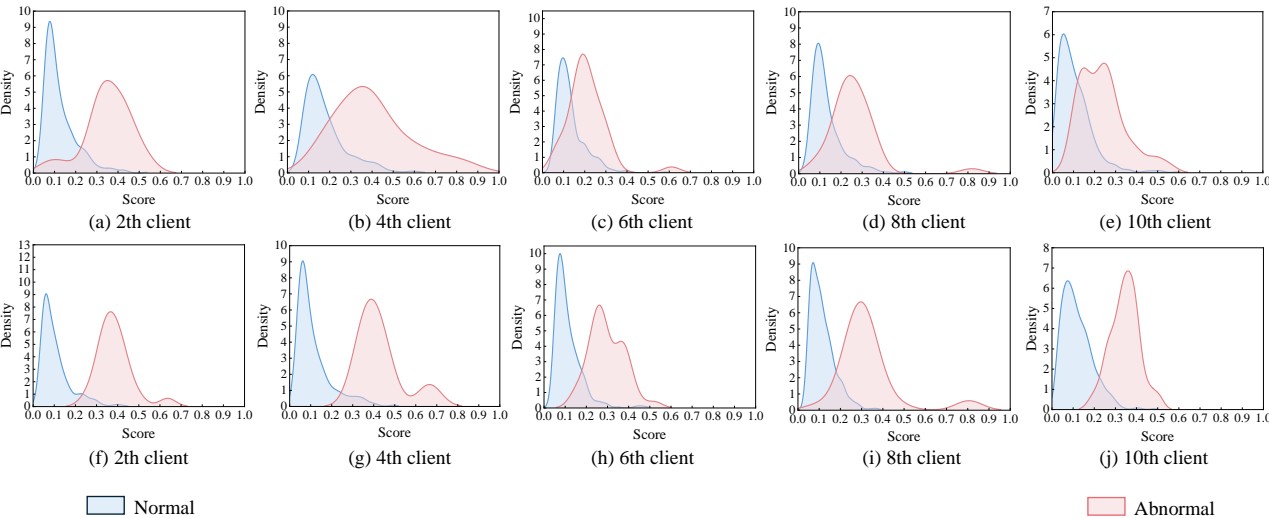

*Figure 14.* Case study on modeling weak anomalies vs. ignoring weak anomalies. The first row shows the results without modeling weak anomalies, while the second row shows the results with weak anomaly modeling.

*Figure 15.* Case study for multi-view learning. The first row shows the results of attribute-aware anomaly learning, the second row shows the results of structure-aware anomaly learning, the third row shows the results of weak anomaly-aware dynamic learning, and the fourth row shows the results of multi-view learning.

score distributions learned by different views vary substantially due to the heterogeneous presence of strong attribute anomalies, strong structural anomalies, and weak anomalies. This highlights the challenge of detecting complex anomaly patterns in federated GAD. 2) No single view can consistently maximize the density peaks of normal and abnormal node score distributions across all client subgraphs, indicating its inability to effectively characterize normal and abnormal nodes. This observation underscores the necessity of multi-view learning. 3) By jointly considering multiple anomaly patterns, MV-FGAD effectively maximizes the density peaks of normal and abnormal node score distributions while minimizing their overlap across client subgraphs. This demonstrates that MV-FGAD can more clearly delineate the distribution boundaries between normal and abnormal nodes, achieving efficient and effective federated GAD.

