# OpenReview forum: "MV-FGAD: Towards Efficient and Effective Federated Graph Anomaly Detection via Multi-view Learning"
_ICML.cc/2026/Conference — ICML 2026 spotlight_

### Official Review · Reviewer_sFU6 · 2026-02-25

**Soundness:** 3
**Presentation:** 3
**Significance:** 3
**Originality:** 3
**Overall Recommendation:** 5
**Confidence:** 4

**Summary:**

This paper studies federated graph anomaly detection (GAD) and identifies a key limitation in existing methods, namely their over-reliance on neighborhood aggregation and their failure to effectively capture weak anomalies in federated settings.

The authors propose MV-FGAD, which consists of: (1) Federated Knowledge Learning via FedAvg-based aggregation. (2) Multi-view Learning with three views: attribute-aware anomaly learning, structure-aware anomaly learning, and weak anomaly-aware dynamic learning. (3) Mahalanobis Distance-based anomaly scoring.

Experiments on seven real-world datasets demonstrate superior performance in AUROC and AUPRC compared to existing federated GAD baselines, along with improved runtime efficiency and scalability.

**Compliance With Llm Reviewing Policy:**

Affirmed.

**Final Justification:**

After the authors' rebuttal and further response, my concerns are largely addressed. Therefore, I have raised my score to 5. It would be better to include the reply content in the final version, which will help readers to further understand the design insights of the model.

**Key Questions For Authors:**

Q1. Can weak anomaly difficulty be theoretically characterized?

Q2. How does the method perform under fully unsupervised federated GAD?

Q3. Can the approach be extended to graph transformers?

**Limitations:**

yes

**Strengths And Weaknesses:**

S1. The methodology is technically coherent and well-structured. The weak anomaly bottleneck is empirically illustrated.

S2. Extensive experiments across datasets and scalability settings. Detailed ablation and runtime analysis are included.

S3. Federated anomaly detection is practically important. The proposed multi-view framework may inspire related federated graph tasks.

---

W1. No evaluation under a fully unsupervised federated GAD, which is also an important GAD scenario.

W2. Mahalanobis distance depends on covariance estimation and matrix inversion. In high-dimensional or small-sample settings, covariance estimation may be unstable, potentially requiring shrinkage or diagonal approximations. The paper does not discuss numerical stability considerations or alternative covariance regularization strategies.

W3. Although the paper emphasizes runtime efficiency, federated learning settings are often more sensitive to communication rounds, parameter upload/download size, bandwidth usage, client dropout, and asynchronous updates. The current analysis mainly focuses on training time rather than communication cost.

W4. Although the paper includes reproducible federated GAD baselines, experiments on additional strong centralized GAD models as upper bounds should be conducted.

---

> ### Author Rebuttal · Authors · 2026-03-30
>
> We sincerely thank **Reviewer sFU6** for the constructive comments.
>
> ---
>
> >**Q1: Can weak anomaly difficulty be theoretically characterized?**
>
> We formalize weak anomaly difficulty ($D_w$) as the statistical homogeneity between normal ($P_N$) and anomalous ($P_A$) distributions: $D_w \propto \int_{s < \tau} P_A(s) ds$, where $\tau$ is the threshold.
>
> **Fig. 3a:** High $D_w$ occurs when anomaly density $P_A(s)$ concentrates within the normal score range ($s < \tau$), rendering them indistinguishable.
>
> **Fig. 3b:** MV-FGAD minimizes $D_w$ by maximizing structural divergence, shifting $P_A$ to the high-score tail to enable separation.
>
> >**Q2: How does the method perform under fully unsupervised federated GAD?**
>
> 1. **Design:** We extend MV-FGAD to the fully unsupervised setting by reformulating the training objective as multi-view consistency learning. Specifically, we employ the MSE loss to minimize the pairwise discrepancies among the representations from the attribute, structural and weak-anomaly views.
> 2. **Analysis:** Table D1 shows that the unsupervised variant matches or surpasses the semi-supervised performance by mitigating local supervision bias from sparse and non-IID labels, while still exhibiting strong potential despite limited generalization.
>
> **Table D1.** GAD performance. “Unsup” and "Semi" indicate unsupervised and semi-supervised learning, respectively.
> |Performance|Model|Reddit|Tolokers|Amazon|Amazon-all|YelpChi|
> |-|-|-|-|-|-|-|
> | AUROC|MV-FGAD*(Unsup)|60.87±0.61|50.47±0.47|78.77±0.85|91.41±0.59|62.91±0.62|
> || MV-FGAD(Semi)|59.27±1.03| 55.73±1.01|78.33±0.74|90.92±0.77|64.25±0.28|
> | AUPRC|MV-FGAD*(Unsup)|6.12±0.59|21.85±0.23|22.67±1.09|40.03±3.44|21.35±0.56|
> ||MV-FGAD(Semi)|5.54±0.51|23.56±0.12|21.91±1.63|40.53±1.74|22.60±0.43|
>
> >**Q3: Can the approach be extended to graph transformers?**
>
> MV-FGAD can be directly extended to Graph Transformers (GT) by utilizing them as the encoders for each view.
>
> **Advantage:** The global attention mechanism of GT can capture global anomalies that local GNNs may miss, enhancing effectiveness.
>
> **Challenge:** GT incurs higher computational cost on large-scale graphs, which may make it difficult for the model to maintain its current efficiency.
>
> >**W1: Please see Q2.**
>
> >**W2: Mahalanobis distance may be unstable in high-dimensional or small-sample settings, with no discussion of numerical stability or regularization.**
>
> 1. **Pseudo-inverse:** MV-FGAD originally employs the pseudo-inverse to handle potential matrix singularity in small-sample federated partitions.
> 2. **Regularization:** Following your suggestion, we further integrate a diagonal perturbation $\lambda\mathbf{I}$ ($\lambda = 10^{-6}$) to ensure strict positive definiteness.
> 3. **Results:** Table D2 shows that these measures ensure numerical stability in extreme settings while preserving the original detection performance.
>
> **Table D2.** GAD performance. “St” indicates stability strategies.
> |Performance|Model|Reddit|Tolokers|Amazon|Amazon-all|YelpChi|
> |-|-|-|-|-|-|-|
> | AUROC| MV-FGAD*(St)|59.83±0.05|54.29±1.04|77.67±0.69|90.74±0.29|64.97±0.83|
> || MV-FGAD(ours)|59.27±1.03| 55.73±1.01|78.33±0.74|90.92±0.77|64.25±0.28|
> | AUPRC|MV-FGAD*(St)|5.57±0.46|23.23±0.53|21.92±0.43|40.10±2.20|23.40±0.71|
> ||MV-FGAD(ours)|5.54±0.51|23.56±0.12|21.91±1.63|40.53±1.74|22.60±0.43|
>
> >**W3: The efficiency analysis mainly focuses on training time rather than communication cost.**
>
> Table D3 reports AUROC on Amazon-all across communication rounds and parameter upload/download sizes. MV-FGAD achieves strong performance within 10 rounds, remains stable as rounds increase, and has the smallest parameter size.
>
> **Table D3.** Communication efficiency analysis.
> |Rounds|LG-FGAD|FGAD|FedCLGN|MV-FGAD(ours)|
> |-|-|-|-|-|
> |10|65.34±16.14|60.41±1.57|48.55±0.86|90.45±0.23|
> |30|62.02±21.08|59.86±3.70|49.19±0.77|90.45±0.03|
> |50|76.36±0.22|59.35±1.14|52.28±0.67|90.80±0.30|
> |70|63.32±1.51|61.28±1.00|55.10±0.88|90.55±0.32|
> |100|57.35±0.52|64.67±0.73|56.10±0.06|90.92±0.77|
> |**Parameter upload/download size (MB)**|0.150|0.126|0.022|0.022|
>
> >**W4: Lack of strong centralized GAD baselines as upper bounds.**
>
> We compare two competitive semi-supervised centralized GAD baselines using the same dataset splits as MV-FGAD. The results are shown in Table D4.
>
> **Table D4.** Centralized GAD performance.
> |Performance|Model|Reddit|Tolokers|Amazon|Amazon-all|YelpChi|
> |-|-|-|-|-|-|-|
> | AUROC|GGAD[1]|62.48±1.33|54.00±2.36|77.54±3.02|92.82±3.37|64.28±3.46|
> || RHO[2]|61.33±2.52|60.17±3.03|79.83±3.94|92.35±1.45|OOM|
> || MV-FGAD(ours)|59.27±1.03| 55.73±1.01|78.33±0.74|90.92±0.77|64.25±0.28|
> | AUPRC|GGAD[1]|5.95±0.81|23.77±1.14|22.72±2.28|64.84±15.06|23.44±1.10|
> || RHO[2]|6.15±0.12|29.12±1.26|22.50±0.94|64.40±2.89|OOM|
> ||MV-FGAD(ours)|5.54±0.51|23.56±0.12|21.91±1.63|40.53±1.74|22.60±0.43|
>
> • [1] Generative semi-supervised graph anomaly detection, 2024.
> • [2] Semi-supervised graph anomaly detection via robust homophily learning, 2025.

---

> > ### Author Rebuttal · Reviewer_sFU6 · 2026-04-01
> >
> > The rebuttal addresses most of my concerns, particularly by providing additional experiments for the unsupervised setting, numerical stability of Mahalanobis distance, communication efficiency, and stronger centralized baselines. These significantly improve the empirical support of the paper.
> > However, the theoretical characterization of weak anomaly difficulty remains largely intuitive without formal guarantees, and the discussion on extending to graph transformers is relatively high-level.
> > Overall, the concerns are largely mitigated, though some aspects that require more experiments remain insufficiently explored. I will maintain my current positive score. If the authors can further address my remaining questions, I will further raise my score.
> >
> > Update: After the authors' further response, my remaining concerns can be largely addressed.

---

> > > ### Author Response · Authors · 2026-04-02
> > >
> > > We sincerely thank **Reviewer sFU6** for the constructive comments and valuable efforts to improve the quality of this work. Due to character limits in our initial response, we were unable to provide the comprehensive discussion. After further analysis and experiments, we offer the following detailed responses to address the concerns.
> > >
> > > ---
> > >
> > > >**1. Can the approach be extended to graph transformers?**
> > >
> > > MV-FGAD can be directly extended to Graph Transformers (GT) by using them as encoders for each view. In practice, we replace the original GCN with the GT, shifting the feature extraction paradigm from local, static neighborhood aggregation to a dynamic self-attention mechanism. Based on the experimental results in Table D5, we observe the following:
> > >
> > > (1) **Enhanced Detection with Robustness Trade-offs:** The GT-based MV-FGAD significantly strengthens the model’s capacity to capture long-range dependencies and non-linear anomalous patterns, outperforming the GCN-based version across most datasets. This shows the strong potential and transferability of the MV-FGAD framework. However, its robustness remains a challenge. For example, the standard deviation of AUPRC on the Amazon dataset reaches 9.93, indicating a heightened sensitivity to subgraph topological variance and noise in high-parameter spaces.
> > >
> > > (2) **Increased Communication Overhead:** The GT-based variant incurs substantially higher communication costs, with parameter upload/download sizes nearly 10$\times$ larger than the original MV-FGAD. This stems from the introduction of multiple learnable matrices.
> > >
> > > (3) **Reduced Computational Efficiency:** GT-based MV-FGAD exhibits longer runtimes across all datasets, a gap that widens as the graph scale increases. This latency is primarily driven by the $O(N^2)$ complexity of attention matrix computations, presenting a trade-off between detection performance and computational efficiency.
> > >
> > > (4) **Conclusion:** In summary, while GT-based encoders bolster the detection performance of MV-FGAD, they introduce non-trivial communication and computational overhead, which contradicts our goal of designing an efficient and effective federated GAD model. Nevertheless, optimizing these high-capacity encoders for federated GAD remains a promising direction for future exploration.
> > >
> > > **Table D5.** GAD performance. “GT” indicates Graph Transformers.
> > > |Performance|Model|Reddit|Tolokers|Amazon|Amazon-all|YelpChi|
> > > |-|-|-|-|-|-|-|
> > > | AUROC| MV-FGAD*(GT)|59.60±0.55|57.38±0.92|83.13±3.79|90.27±1.52|67.65±0.39|
> > > || MV-FGAD(ours)|59.27±1.03| 55.73±1.01|78.33±0.74|90.92±0.77|64.25±0.28|
> > > | AUPRC|MV-FGAD*(GT)|5.55±0.12|25.02±0.19|38.63±9.93|44.48±2.01|28.07±0.62|
> > > ||MV-FGAD(ours)|5.54±0.51|23.56±0.12|21.91±1.63|40.53±1.74|22.60±0.43|
> > > | Running Time (s) |MV-FGAD*(GT)|35.24|33.01|33.39|44.19|145.44|
> > > ||MV-FGAD(ours)|31.74|32.94|33.32|34.64|104.51|
> > > |Parameter upload/download size (MB)|MV-FGAD*(GT)|0.240|0.203|0.217|0.217|0.224|
> > > ||MV-FGAD(ours)|0.031|0.010|0.022|0.022|0.024|
> > >
> > > >**2. Can weak anomaly difficulty be theoretically characterized?**
> > >
> > > (1) **Formalizing $D_w$:** We define weak anomaly difficulty $D_w \propto \int_{s < \tau} P_A(s) ds$, as the mass of the anomalous distribution $P_A$ overlapping with the normal distribution $P_N$. The overlap is quantified by the Total Variation (TV) distance: $d_{TV}(P_N, P_A) = 1 - \int \min(p_N(s), p_A(s)) ds$. As $D_w \to 1$, the overlap integral $\int \min(p_N, p_A)$ grows, forcing $d_{TV} \to 0$.
> > >
> > > (2) **Theoretical Lower Bound:** According to Le Cam’s Lemma, for any detector $\Psi$, the minimum average error probability $P_e$ is bounded by: $P_e \ge \frac{1}{2} (1 - d_{TV}(P_N, P_A))$. When $D_w \to 1$, then $d_{TV} \to 0$, implying $P_e \to 0.5$ (random guessing). This proves that high $D_w$ creates a theoretical detection bottleneck where anomalies become statistically indistinguishable from normal nodes.
> > >
> > > (3) **Mitigation via MV-FGAD:** Our framework addresses this bottleneck through two theoretically grounded mechanisms: 1) While our objective $\mathcal{L} _ m$ (Eq. 21) is formulated via MHD-based Loss, it serves as a functional proxy to maximize $d _ {TV}$. By minimizing the variance of normal representations and the abnormal representations to enforce a separation margin, MV-FGAD reconfigures the latent embedding to induce a larger distributional divergence. 2) According to Pinsker’s Inequality ($d _ {TV} \le \sqrt{\frac{1}{2} D _ {KL}}$), our multi-view learning acts as a constrained optimization that pushes the lower bound of $d_{TV}$ higher. By capturing complementary signals across strong attribute, strong structure, and weak anomaly views, MV-FGAD ensures a non-zero divergence for weak anomalies, reducing $D_w$ and shifting $P_A$ to the high-score tail.

---

### Official Review · Reviewer_5oru · 2026-02-27

**Soundness:** 3
**Presentation:** 4
**Significance:** 3
**Originality:** 3
**Overall Recommendation:** 4
**Confidence:** 4

**Summary:**

This paper focuses on federated graph anomaly detection. The authors first identify two key limitations of existing methods: their over-reliance on neighborhood aggregation under partitioned federated settings and their inability to effectively detect weak attribute or structural anomalies. To address these issues, they propose MV-FGAD, a multi-view federated framework that integrates federated knowledge learning and structure optimization to mine anomalies of varying strengths.

**Compliance With Llm Reviewing Policy:**

Affirmed.

**Key Questions For Authors:**

Please refer to the Weaknesses.

**Limitations:**

Yes.

**Strengths And Weaknesses:**

Strengths

1. This paper draws attention to weak anomalies in GAD, and appears to be the first study in the GAD field to explicitly investigate this issue.
2. This paper proposes an efficient and effective framework to address two key limitations of existing federated GAD methods.

Weaknesses

1. Mahalanobis distance is a classic metric algorithm, but it is rarely used in GAD. Although the authors conducted ablation experiments to prove its effectiveness, they still need to explain and prove why they chose it as the anomaly score metric for MV-FGAD, rather than other anomaly detection algorithms.
2. The federated GAD baselines used are somewhat limited, and including all five existing federated GAD models described in the related work as comparison baselines would provide a more comprehensive evaluation.
3. The term "semi-supervised federated GAD" in the Problem Statement is inappropriate because Eq. (18) states that the model uses all the labels from the training set.
4. Although the overall structure of the paper is clear, some parts are difficult to understand. For example, in the model training section, it is unclear how Eq. (21) is derived.
5. Section 5.1 introduces seven datasets, but why do only five datasets appear in the Superiority Results section?

---

> ### Author Rebuttal · Authors · 2026-03-30
>
> We sincerely thank **Reviewer 5oru** for the constructive comments.
>
> ---
>
> >**W1: Why choose Mahalanobis distance over other anomaly scoring methods?**
> 1. **Multi-view Feature Fusion:** Unlike Euclidean distance and cosine similarity, Mahalanobis distance (MHD) incorporates the covariance matrix to achieve automatic feature decoupling and normalization in the feature space, effectively reducing redundancy across multiple views.
> 2. **Anomaly Awareness:** MHD is highly sensitive to boundary anomalies in non-spherical distributions.
> 3. **Experiments:** As shown in Table C1, MHD outperforms Euclidean distance and cosine similarity in detection performance.
>
> **Table C1.** Ablation study for anomaly scoring.
> |Performance|Model|Reddit|Tolokers|Amazon|Amazon-all|YelpChi|
> |-|-|-|-|-|-|-|
> | AUROC|Euclidean Distance|56.79±1.49|54.67±1.46|71.11±0.99|71.05±2.00|60.83±0.08|
> ||Cosine Similarity|55.66±1.43|55.20±0.87|72.19±0.18|71.13±2.31|60.84±0.41|
> ||MHD|59.27±1.03| 55.73±1.01|78.33±0.74|90.92±0.77|64.25±0.28|
> | AUPRC|Euclidean Distance|5.02±1.44|23.17±1.24|20.83±1.33|19.04±3.88|20.85±0.49|
> ||Cosine Similarity|5.16±0.21|23.31±0.41|20.85±1.38|12.69±1.43|20.98±0.34|
> ||MHD|5.54±0.51|23.56±0.12|21.91±1.63|40.53±1.74|22.60±0.43|
> >**W2: Please include all five existing federated GAD models mentioned in the related work as comparison baselines for a comprehensive evaluation.**
>
> We plan to consider all five federalized GAD models mentioned in the related work as baselines. However, two lack publicly available code, so we use only the remaining three to ensure reproducibility and fairness.
>
> >**W3: The term "semi-supervised federated GAD" in the Problem Statement is inappropriate because Eq. (18) states that the model uses all the labels from the training set.**
>
> We apologize for the lack of clarity in our initial description of the training labels.
>
> 1. **Explanation:** We would like to clarify that only 20\% of nodes in the global graph are labeled ($|\mathcal{V} _ {l}| \ll |\mathcal{V} _ {total}|$) in MV-FGAD. While Eq. (18) utilizes the available labels $\mathcal{V}_l$ within each client’s local training set to guide the optimization, the model must leverage the GCN to propagate information and capture the representations of the entire graph, including the vast majority of unlabeled nodes.
>
> 2. **Definition:** In graph learning [1], a setting where a model is trained on a limited labeled subset to generalize over a large unlabeled graph structure is strictly defined as semi-supervised learning, distinct from fully supervised learning where every node's ground truth is known during training.
>
> 3. **Action:** If we have the opportunity to submit the camera-ready version, we will incorporate this explanation into the Model Training Section.
>
> • [1] Graph-based semi-supervised learning: A comprehensive review, 2022.
>
> >**W4: In the model training section, it is unclear how Eq. (21) is derived.**
>
> 1. **Derivation:** Eq. (21) is a joint loss function designed to integrate knowledge from three complementary perspectives. Specifically, $\mathcal{L}_a$, $\mathcal{L}_s$ and $\mathcal{L}_w$ are instances of the loss defined in Eq. (20), where the same optimization objective ($\mathcal{L} _ f=\mathcal{L} _ {nor}+\mathcal{L} _ {ano}$) is applied to the attribute view, structural view and weak-anomaly view, respectively.
>
> 2. **Motivation:** The final loss $\mathcal{L}_m$ is the summation of these view-specific losses. This joint optimization ensures that the GCN captures diverse anomaly patterns across multiple views simultaneously, leading to more robust detection.
>
> 3. **Action:** If we have the opportunity to submit a camera-ready version, we will incorporate this explanation into the Model Training Section.
>
> >**W5: Section 5.1 introduces seven datasets, but why do only five datasets appear in the Superiority Results section?**
>
> We utilize a total of seven datasets to evaluate MV-FGAD. To ensure the comprehensiveness of our experiments, we partition the experiments as follows:
>
> 1. **Overall Effectiveness**: Five widely-used GAD datasets are discussed in the Superiority Results Section to validate the overall effectiveness of the model.
>
> 2. **Scalability & Robustness**: Two large-scale datasets are specifically analyzed in the Scalability Analysis Section to demonstrate the model’s efficiency and capability in handling large-scale graph data.

---

> > ### Author Rebuttal · Reviewer_5oru · 2026-04-02
> >
> > The authors have solved my problems and I will keep my positive score.

---

### Official Review · Reviewer_WWeB · 2026-03-09

**Soundness:** 4
**Presentation:** 4
**Significance:** 3
**Originality:** 3
**Overall Recommendation:** 5
**Confidence:** 4

**Summary:**

To address the detection bottleneck in partitioned subgraphs, this paper proposes MV-FGAD, a multi-view federated framework that optimizes local topologies through collaborative knowledge learning and employs Mahalanobis distance-based scoring to effectively quantify diverse anomaly patterns.

**Compliance With Llm Reviewing Policy:**

Affirmed.

**Final Justification:**

The authors did a good job with the responses, and I think all my previous concerns have been addressed. I will maintain my positive judgment.

**Key Questions For Authors:**

Please see my above W1-W6.

**Limitations:**

yes

**Strengths And Weaknesses:**

**Strengths**:

S1. This paper provides a pioneering systematic analysis of anomaly strength in federated GAD, moving beyond conventional neighborhood aggregation to effectively capture "weak" anomaly signals that are often overlooked in partitioned subgraphs.

S2. MV-FGAD introduces a robust multi-view learning mechanism and Mahalanobis distance scoring, demonstrating superior efficiency and effectiveness in quantifying diverse anomaly patterns across multiple real-world benchmarks.

**Weaknesses**:

W1. I am curious why the final anomaly score is computed as the average across different views rather than taking the maximum score for each node. The authors should provide further explanation and experimental justification for this design choice.

W2. This paper explains that weak anomalies are the cause of overlap between normal and abnormal data. But as long as the model's detection performance is poor, this overlap will occur regardless of the type of anomaly.

W3. Since higher-order propagation in Eq. (9) may cause over-smoothing, how does the model prevent structural anomaly signals from being weakened as $k$ increases?

W4. Since both $H_c^{(l)}$ and $H_d^{(l)}$ are derived from the same relational matrix, does their decomposition truly introduce complementary information, or is it essentially a reparameterization of the same signal?

W5. The propagation in Eq. (13) accumulates relational similarity signals, but could this process also smooth weak anomalies and reinforce dominant normal patterns?

W6. Please provide a detailed analysis of which aspects of MV-FGAD’s design contribute to its efficiency, and what insights this offers for the development of the GAD community.

---

> ### Author Rebuttal · Authors · 2026-03-30
>
> We sincerely thank **Reviewer WWeB** for the constructive comments.
>
> ---
>
> >**W1: Why average anomaly scores across views instead of using the maximum?**
>
> 1. **Complementarity:** The confidence scores provided by different views are complementary. Averaging effectively acts as an ensemble strategy, which is known to be more stable than selecting a single maximum value in machine learning.
>
> 2. **Robustness:** Unlike the maximum strategy, which is prone to single-view noise, averaging better preserves subtle signals across multiple views. This ensures that weak anomalies are not suppressed by dominant noise.
>
> 3. **Experiments:** As supported by the analysis above, the results in Table B1 further demonstrate that, although the maximum strategy is a plausible design, it is less effective than averaging in comprehensively capturing anomaly information.
>
> **Table B1.** Ablation study for multi-view computing.
> |Performance|Model|Reddit|Tolokers|Amazon|Amazon-all|YelpChi|
> |-|-|-|-|-|-|-|
> | AUROC|MV-FGAD*(Max)|60.28±1.08|51.32±0.63|78.17±0.11|89.02±1.23|63.49±0.88|
> ||MV-FGAD(Average)|59.27±1.03| 55.73±1.01|78.33±0.74|90.92±0.77|64.25±0.28|
> | AUPRC|MV-FGAD*(Max)|4.98±0.15|21.71±0.33|22.38±0.34|33.94±0.83|21.67±0.89|
> ||MV-FGAD(Average)|5.54±0.51|23.56±0.12|21.91±1.63|40.53±1.74|22.60±0.43|
>
> >**W2: Weak anomalies are claimed to cause overlap, but poor model performance can also lead to such overlap.**
>
> We would like to clarify that "overlap" and "poor performance" are not independent issues, but rather a cause-and-effect relationship.
>
> 1. **Theory:** As defined in our response to reviewer sFU6's Q1, weak anomalies are characterized by high statistical homogeneity with normal data. This intrinsic feature-space overlap exists prior to inference and is likely to lead to poor performance.
>
> 2. **Experiments:** As shown in Fig. 5 and Fig. 12 of the paper, significant overlap exists between distributions without Weak Anomaly-aware Dynamic Learning (WADL), the overlap is significant. By introducing WADL, the anomalous distribution $P_A$ is effectively shifted toward the high-score tail, showing that our model reshapes the latent space to reveal hidden patterns and reduce the weak anomaly $D_w$.
>
> >**W3: How does the model prevent over-smoothing weakening anomalies as $k$ increases?**
>
> 1. **Low-order Locality:** We set $k=2$ to capture essential higher-order topological correlations (2-hop paths) while avoiding the depth ($k \ge 4$) where over-smoothing typically occurs. This ensures that structural anomaly signals remain localized and distinct.
>
> 2. **Multi-scale Information:** As shown in Eq. (9), we concatenate representations from different scales: $\widetilde{\mathbf{X}} _ {con} = [\widetilde{\mathbf{X}} ^ {(1)}, \dots, \widetilde{\mathbf{X}} ^ {(k)}]$. By integrating low-order signals (which are highly sensitive to anomalies) with higher-order information, this design prevents the "averaging" effect of deep propagation and maintains the discriminative power of the structural representation.
>
> >**W4: Do $\mathbf{H}^{(l)} _ {c}$ and $\mathbf{H}^{(l)} _ {d}$ provide complementary information or just reparameterize the same signal?**
>
> 1. **Non-linear Disentanglement:** Although $\mathbf{H}^{(l)} _ {c}$ and $\mathbf{H}^{(l)} _ {d}$ both originate from $\widetilde{\mathbf{S}}^{(l)}$, we apply polarized activations $\sigma(\widetilde{\mathbf{S}}^{(l)})$ and $\sigma(-\widetilde{\mathbf{S}}^{(l)})$. Since $\sigma$ (PReLU) is asymmetric and non-linear, these filters act as antagonistic masks that selectively amplify normal consistency and anomalous patterns, respectively.
>
> 2. **Semantic Complementarity:** $\mathbf{H}^{(l)} _ {c}$ functions as a low-pass filter to reinforce stable global structures, whereas $\mathbf{H}^{(l)}_{d}$ acts as a high-pass filter to isolate local discordance.
>
> >**W5: Does the propagation in Eq. (13) oversmooth weak anomalies while reinforcing dominant normal patterns?**
>
> We would like to clarify that Eq. (13) amplifies weak anomalies rather than smoothing them.
>
> 1. **Feature Smoothing:** Unlike smoothing node features, we aggregate relational similarity $\mathbf{R}^{(l)}$. This captures layer-wise consistency, allowing subtle deviations to accumulate into prominent signals.
>
> 2. **Contrastive Amplification:** $\beta$ acts as a momentum controller, serving as a denoising mechanism that stabilizes the normal core while highlighting persistent anomalous discordance.
>
> >**W6: What makes MV-FGAD efficient, and what are its implications for GAD community?**
>
> 1. **Design:** By specifically designing attribute, structural, and weak-anomaly view learning, MV-FGAD eliminates redundant computations inherent in traditional GAD models. Combined with Mahalanobis distance (MHD) scoring, it enables more efficient GAD.
>
> 2. **Implications:** Multi-view learning tailored for anomaly detection can more effectively capture anomalous signals, providing an efficient and effective paradigm for federated GAD.

---

> > ### Author Rebuttal · Reviewer_WWeB · 2026-04-03
> >
> > The authors did a good job with the responses, and I think all my previous concerns have been addressed. I will maintain my positive judgment.

---

### Official Review · Reviewer_hNeX · 2026-03-12

**Soundness:** 3
**Presentation:** 4
**Significance:** 3
**Originality:** 3
**Overall Recommendation:** 5
**Confidence:** 5

**Summary:**

This paper revisits federated GAD and highlights the challenges of partitioned subgraphs and weak anomalies. It proposes MV-FGAD, a multi-view federated framework that enhances anomaly detection through shared knowledge learning and cross-view scoring, achieving strong experimental performance.

**Compliance With Llm Reviewing Policy:**

Affirmed.

**Final Justification:**

The rebuttal has well addressed my concerns.

**Key Questions For Authors:**

The important points listed in weakness 1-5.

**Limitations:**

yes

**Strengths And Weaknesses:**

**Strengths**

1. MV-FGAD effectively captures weak and strong anomaly signals through multi-view learning, leading to more accurate and comprehensive detection.
2. MV-FGAD demonstrates high efficiency and effectiveness across diverse federated settings and datasets.

**Weaknesses**

1. MV-FGAD involves federated knowledge learning, multi-view modeling, and MHD-based scoring. How does the computational and communication overhead scale with the number of views and clients, and how does it compare to existing federated GAD methods in large-scale settings?
2. How does MV-FGAD ensure that different views provide complementary rather than redundant information, especially under incomplete local neighborhoods in federated settings?
3. While decoupling attribute learning from neighborhood aggregation prevents anomaly "obscuring", does this approach risk overlooking contextual anomalies that are only identifiable through their inconsistency with local surroundings?
4. Since probabilistic propagation matrix is derived from federated knowledge, how does the model handle potential "noise" or "biases" transferred from other clients that might have very different topological characteristics?
5. From Table 1, LG-FGAD shows large standard deviations on certain datasets, indicating potential instability. Please analyze the reasons behind these stability differences and clarify which design choices in MV-FGAD contribute to its more consistent performance in federated settings.

---

> ### Author Rebuttal · Authors · 2026-03-30
>
> We sincerely thank **Reviewer hNeX** for the constructive comments.
>
> ---
>
> >**W1: How do the computational and communication costs of MV-FGAD scale with the number of views and clients, and how do they compare to existing federated GAD methods?**
>
> 1. **Computational and Communication Costs:** Since all views share the single GCN backbone, each client performs $V$ forward passes (one per view), leading to computational complexity of $O(V \times \text{C}_{\text{GCN}})$. This cost is independent of the total number of clients $N$. Globally, the shared-parameter architecture ensures that the model size $|\theta|$ does not grow with the number of views. Consequently, the per-round communication overhead remains $O(N \times |\theta|)$, which is as efficient as vanilla single-view federated GAD. In large-scale settings, this linear scaling with $N$ can be further optimized via client sampling.
>
> 2. **Comparison in Large-Scale Settings:** Existing federated GAD methods often struggle with "weak anomaly" signals and struggle to balance model expressiveness with communication costs. MV-FGAD captures complex multi-view patterns for significantly higher detection performance without exploding the parameter space.
>
> >**W2: How does MV-FGAD ensure view complementarity rather than redundancy, especially with incomplete local neighborhoods?**
>
> 1. **To prevent redundancy:** MV-FGAD enforces strict architectural constraints by restricting the attribute-aware view to MLP-based extraction (Eq. 8) to bypass neighbor smoothing. Simultaneously, the weak anomaly-aware view utilizes a dual-signal decomposition (Eq. 14) to explicitly separate deviation signals from consistency patterns, ensuring each view targets distinct anomaly categories.
>
> 2. **To ensure complementarity under incomplete data:** MV-FGAD utilizes aggregated global model weights to infer potential nodal relationships in the latent space (Eq. 6). This allows clients to compensate for sparse local topologies using shared global knowledge without compromising data privacy.
>
> >**W3: Does decoupling attribute learning from neighborhood aggregation risk missing contextual anomalies?**
>
> 1. **Attribute-aware View:** The attribute-aware view bypasses neighborhood aggregation to isolate strong attribute anomalies often obscured by GNN smoothing. It is not intended for contextual outliers but to enhance sensitivity to attribute-level deviations.
>
> 2. **Contextual Anomaly Capture:** Contextual anomalies are explicitly targeted by the structure-aware (Eq. 9 and Eq. 10) and weak anomaly-aware views (Eq. 15). While the former identifies strong contextual discrepancies using topological and attribute information, the latter employs the deviation signal $\mathbf{H}_d$ to quantify subtle relational misalignments in the semantic neighborhood.
>
> 3. **Multi-view Scoring:** The final score $s_i$ aggregates all outputs in Eq. 17. A contextual anomaly may appear normal in the attribute view but will trigger significant MHD deviations in the structure-aware or weak anomaly-aware views, ensuring robust detection across diverse anomaly patterns.
>
> >**W4: How does the model mitigate noise or bias in the federated propagation matrix from heterogeneous clients?**
>
> 1. **Adaptive Fusion:** MV-FGAD uses hyperparameter $\alpha$ to balance local topology $\mathbf{A}_i$ with global knowledge $\hat{\mathbf{P}}_i$ in Eq. 6, preventing the local model from being overwhelmed by global noise or bias.
>
> 2. **Multi-view Learning:** Since the three views are functionally decoupled, it is statistically improbable for noise or bias to simultaneously corrupt all representation spaces. Aggregating these independent scores dilutes the impact of localized noise, providing a more stable detection than single-view approaches.
>
> >**W5: Why does LG-FGAD exhibit high variance, and which design choices make MV-FGAD more stable in federated settings?**
>
> 1. **LG-FGAD Instability:** The instability of LG-FGAD stems from its adversarial training paradigm, where the quality of self-generated anomalies is highly sensitive to local distributions, leading to inconsistent gradient updates. Additionally, the dual distillation process can incur knowledge drift when distilling complex global patterns into lightweight student models under data sparsity.
>
> 2. **MV-FGAD Stability:** 1) Stable prior: MV-FGAD uses federated knowledge transfer to provide a stable prior, leveraging global insights to ensure consistent representation learning even for clients with incomplete local topologies. 2) Multi-view learning: The multi-view mechanism decouples representations from different perspectives to comprehensively capture diverse anomaly patterns (strong attribute, strong structural and weak anomalies), thereby reducing local noise. 3) Anomaly scoring: The local MHD-based scoring integrates information from multiple views, effectively suppressing variance caused by data heterogeneity across clients.

---

> > ### Author Rebuttal · Reviewer_hNeX · 2026-04-02
> >
> > The response has addressed my concerns.

---

### Decision · Program_Chairs · 2026-04-30

**Decision:**

Accept (spotlight)

**Comment:**

This paper studies the problem of anomaly detection in federated graphs, and use a multi-view learning strategy combined with federated knowledge sharing to effectively alleviate the problem that weak anomalies in distributed subgraphs are difficult to detect. Extensive experiments verify the effectiveness of our method. The reviewer raised concerns regarding multi-view redundancy, computational/communication overhead, soundness of the definition of weak anomalies, and numerical stability of Mahalanobis distance. However, the authors addressed the reviewer's concerns one by one in rebuttal, and all reviewers showed positive evaluations.